# STANDARD GAUSSIAN PROCESS IS ALL YOU NEED FOR HIGH-DIMENSIONAL BAYESIAN OPTIMIZATION

**Zhitong Xu, Haitao Wang, Jeff M. Phillips, Shandian Zhe**[*]
Kahlert School of Computing
University of Utah, Salt Lake City, UT 84112, USA
`{u1502956, haitao.wang}@utah.edu, {jeffp, zhe}@cs.utah.edu`

## ABSTRACT

A long-standing belief holds that Bayesian Optimization (BO) with standard Gaussian processes (GP) — referred to as standard BO — underperforms in high-dimensional optimization problems. While this belief seems plausible, it lacks both robust empirical evidence and theoretical justification. To address this gap, we present a systematic investigation. First, through a comprehensive evaluation across twelve benchmarks, we found that while the popular Square Exponential (SE) kernel often leads to poor performance, using Matérn kernels enables standard BO to consistently achieve top-tier results, frequently surpassing methods specifically designed for high-dimensional optimization. Second, our theoretical analysis reveals that the SE kernel's failure primarily stems from improper initialization of the length-scale parameters, which are commonly used in practice but can cause gradient vanishing in training. We provide a probabilistic bound to characterize this issue, showing that Matérn kernels are less susceptible and can robustly handle much higher dimensions. Third, we propose a simple robust initialization strategy that dramatically improves the performance of the SE kernel, bringing it close to state-of-the-art methods, without requiring additional priors or regularization. We prove another probabilistic bound that demonstrates how the gradient vanishing issue can be effectively mitigated with our method. Our findings advocate for a re-evaluation of standard BO's potential in high-dimensional settings. The code is released at `https://github.com/XZT008/Standard-GP-is-all-you-need-for-HDBO`.

## 1 Introduction

Many applications require optimizing complex functions, allowing queries only for the function values, possibly with noises, yet without any gradient information. Bayesian Optimization (BO) (Snoek et al., 2012; Mockus, 2012) is a popular approach for addressing such challenges. BO typically employs Gaussian process (GP) (Williams & Rasmussen, 2006) as a probabilistic surrogate. It iteratively approximates the target function, integrates the posterior information to maximize an acquisition function for generating new inputs at which to query, then updates the GP model with new examples, and concurrently approach the optimum.

Despite numerous successes, there has been a widespread belief that BO with standard GP regression, referred to as standard BO, is limited to low-dimensional optimization problems (Frazier, 2018; Nayebi et al., 2019; Eriksson & Jankowiak, 2021; Moriconi et al., 2020; Letham et al., 2020; Wang et al., 2016; Li et al., 2016). It is commonly thought that the number of optimization variables should not exceed 15 or 20, as beyond this threshold, BO is prone to failure (Frazier, 2018; Nayebi et al., 2019). A continuous line of research is dedicated to developing novel high-dimensional BO methods. The fundamental strategy is to impose strong structural assumptions into surrogate modeling so as to avoid directly dealing with high-dimensional inputs. One class of methods assumes a decomposition structure within the functional space (Kandasamy et al., 2015; Rolland et al., 2018; Han et al., 2021; Ziomek & Ammar, 2023), where the target function is expressed as the summation of a group of low-dimensional functions, each operating over a small number of variables. Another family of methods assumes that the inputs are intrinsically low-rank. These methods either project the original

---

[*]Corresponding author

inputs into a low-dimensional space (Wang et al., 2016; Nayebi et al., 2019; Letham et al., 2020) or use sparse-inducing priors to trim down massive input variables (Eriksson & Jankowiak, 2021). Subsequently, GP surrogates are built with the reduced input dimensions.

While the aforementioned concerns may seem plausible, there is a lack of both strong empirical evidence and theoretical justification to confirm and explain why standard BO would be ineffective in high-dimensional optimization. To bridge this gap, we systematically investigated standard BO in this paper. The major contributions of our work are summarized as follows:

- **Empirical Results.** We investigated BO with standard GP across eleven widely used benchmarks and one novel benchmark, encompassing six synthetic and six real-world high-dimensional optimization tasks. The number of variables ranged from 30 to 1,003. We compared standard BO with nine state-of-the-art high-dimensional BO methods, and performed extensive evaluations. Surprisingly, while the popular SE kernel often led to poor performance, switching to the ARD Matérn kernel enabled standard BO to nearly always achieve the best (or near-best) optimization performance. When successful, standard BO seems more flexible in accommodating various structures within the target functions.
- **Theory.** Through analyzing the gradient structure of the GP likelihood, we identified the primary failure mode of standard BO, in particular when using the SE kernel. Specifically, the commonly used initializations for the length-scale parameters, such as setting them to one, is improper in high-dimensional settings and can easily cause gradient vanishing, preventing effective training. We proved a probabilistic tail bound to characterize this issue under mild conditions. Applying the same analytical framework to the Matérn kernel, we show through a comparison of the tail bounds that the Matérn kernel is less prone to gradient vanishing with the same length-scale initialization, making it more effective for handling high-dimensional problems.
- **Simple Robust Initialization.** Based on our theoretical results, we proposed a simple yet robust length-scale initialization method, without requiring any additional priors or regularization. We proved a probabilistic bound showing that the probability of gradient vanishing decreases exponentially with the increase in dimensionality, implying that the gradient vanishing issue can be effectively mitigated in high-dimensional settings. Empirical evaluations demonstrate that our initialization method dramatically improves the performance of standard BO with SE kernels, enabling it to achieve, or come close to, the state-of-the-art performance in high-dimensional optimization.

## 2 Standard Bayesian Optimization

Consider maximizing a $d$-dimensional objective function $f : \mathcal{X} \subset \mathbb{R}^d \to \mathbb{R}$, where the function form is unknown. We can query the function value at any input $\mathbf{x} \in \mathcal{X}$, possibly with noises, but no gradient information is accessible. We aim to find $\mathbf{x}^\dagger = \text{argmax}_{\mathbf{x} \in \mathcal{X}} f(\mathbf{x})$. To achieve this, Bayesian Optimization (BO) employs a probabilistic surrogate model to predict $f$ across the domain $\mathcal{X}$, while also quantifying the uncertainty of the prediction. This information is then integrated to compute an acquisition function, which measures the utility of querying at any new input location given the current function estimate. By maximizing the acquisition function, BO identifies a new input location at which to query, ideally closer to the optimum. Concurrently, the acquired new example is incorporated into the training dataset, and the surrogate model is retrained to improve the accuracy. BO begins by querying at a few (typically randomly selected) input locations, and trains an initial surrogate model. The iterative procedure repeats until convergence or a stopping criterion is met.

The standard BO adopts GP regression (Williams & Rasmussen, 2006) for surrogate modeling. A GP prior is placed over the target function, $f \sim \mathcal{GP}\left(m(\mathbf{x}), \kappa(\mathbf{x}, \mathbf{x}')\right)$, where $m(\mathbf{x})$ is the mean function, which is often set as a constant, and $\kappa(\cdot, \cdot)$ is the covariance function, which is usually chosen as a Mercer kernel function. One most popular kernel used in BO is the Square Exponential (SE) kernel with Automatic Relevance Determination (ARD),

$$\kappa_{\text{SE}}(\mathbf{x}, \mathbf{x}') = a \exp(-\rho^2), \tag{1}$$

where $a > 0$ is the amplitude, $\rho = \sqrt{(\mathbf{x} - \mathbf{x}')^\top \text{diag}(\frac{1}{\ell^2})(\mathbf{x} - \mathbf{x}')}$, and $\boldsymbol{\ell} = (\ell_1, \ldots, \ell_d)^\top > \mathbf{0}$ are the length-scale parameters. We refer to this kernel as ARD because each input dimension has a

distinct length-scale parameter. An alternative choice is the ARD Matérn-5/2 kernel,

$$\kappa_{\text{Matérn}}(\mathbf{x}, \mathbf{x}') = a \left(1 + \sqrt{5}\rho + 5\rho^2/3\right) \exp\left(-\sqrt{5}\rho\right). \tag{2}$$

Given training inputs $\mathbf{X} = [\mathbf{x}_1, \ldots, \mathbf{x}_N]^\top$ and (noisy) outputs $\mathbf{y} = [y_1, \ldots, y_N]^\top$, let us denote the function values at $\mathbf{X}$ as $\mathbf{f} = [f(\mathbf{x}_1), \ldots, f(\mathbf{x}_N)]^\top$, which according to the GP prior, follow a multi-variate Gaussian distribution, $p(\mathbf{f}) = \mathcal{N}(\mathbf{f}|\mathbf{m}, \mathbf{K})$, where $\mathbf{m} = [m(\mathbf{x}_1), \ldots, m(\mathbf{x}_N)]^\top$, $\mathbf{K}$ is the kernel matrix on $\mathbf{X}$, and each $[\mathbf{K}]_{ij} = \kappa(\mathbf{x}_i, \mathbf{x}_j)$. Then one can employ a Gaussian likelihood for the observations $\mathbf{y}$, and

$$p(\mathbf{y}|\mathbf{X}) = \mathcal{N}(\mathbf{y}|\mathbf{m}, \mathbf{K} + \sigma^2\mathbf{I}), \tag{3}$$

where $\sigma^2$ is the noise variance. To estimate the GP parameters, *e.g.,* the length-scales $\boldsymbol{\ell}$, one can maximize the marginal likelihood (3). The predictive distribution at any new input $\mathbf{x}^*$, is conditional Gaussian, $p\left(f(\mathbf{x}^*)|\mathbf{y}\right) = \mathcal{N}\left(f(\mathbf{x}^*)|\mu(\mathbf{x}^*), v(\mathbf{x}^*)\right)$, where $\mu(\mathbf{x}^*) = m(\mathbf{x}^*) + \kappa(\mathbf{x}^*, \mathbf{X})(\mathbf{K} + \sigma^2\mathbf{I})^{-1}(\mathbf{y} - \mathbf{m})$, and $v(\mathbf{x}^*) = \kappa(\mathbf{x}^*, \mathbf{x}^*) - \kappa(\mathbf{x}^*, \mathbf{X})(\mathbf{K} + \sigma^2\mathbf{I})^{-1}\kappa(\mathbf{X}, \mathbf{x}^*)$.

In each iteration, given the current GP surrogate model, an acquisition function is maximized to identify the next input location at which to query. One commonly used acquisition function is the upper confidence bound (UCB), defined as $\text{UCB}(\mathbf{x}) = \mu(\mathbf{x}) + \lambda\sqrt{v(\mathbf{x})}, \quad \mathbf{x} \in \mathcal{X}$, where $\lambda$ represents the exploration level. There have been other popular acquisition functions, such as Expected Improvement (EI) (Jones et al., 1998), Thompson sampling (TS) (Russo et al., 2018), the recently proposed log-EI (Ament et al., 2024), among others.

## 3 High Dimensional Bayesian Optimization

An enduring and widespread belief is that when the input dimension $d$ is high, *e.g.,* a few hundreds, the standard BO is prone to failure. This belief might partly arise from an intuition that commonly used kernels, such as (1), could encounter challenges in dealing with high-dimensional inputs, making GP struggle in capturing the target function. A dedicated line of research is developing novel high-dimensional BO methods. The key idea of these methods is to impose some structural assumptions in surrogate modeling to sidestep directly handling the high-dimensional inputs in kernels.

**Structural Assumption in Functional Space.** The first class of methods assumes a decomposition structure within the functional space. Specifically, the target function $f$ is modeled as

$$f(\mathbf{x}) = \sum_{j=1}^{M} f_j(\mathbf{x}^j), \ \ f_j \sim \mathcal{GP}\left(m_j(\mathbf{x}^j), \kappa_j(\cdot, \cdot)\right), \tag{4}$$

where each $\mathbf{x}^j \subset \mathbf{x}$ is a small group of input variables, and $\mathbf{x} = \mathbf{x}^1 \cup \ldots \cup \mathbf{x}^M$. There can be a variety of choices for the group number $M$ and each variable group $\mathbf{x}^j$. In (Kandasamy et al., 2015), the input $\mathbf{x}$ is partitioned into non-overlapped groups. After every a few BO iterations, a new partition is selected from a set of random partitions. The selection is based on the model evidence. In (Rolland et al., 2018), the variable groups can overlap, and are represented as the maximum cliques on a dependency graph. A Gibbs sampling method was developed to learn the structure of the dependency graph. Han et al. (2021) proposed Tree-UCB, which restricts the dependency graph to be tree-structured so as to boost the efficiency of structure learning. However, the more recent work (Ziomek & Ammar, 2023) points out that, learning the group structure through relatively small data can be misleading. A wrongly learned structure can cause queries to be stuck at local optima. This work proposes RDUCB that randomly decomposes $\mathbf{x}$ into a tree structure, and surprisingly works better than those learned structures through various methods.

**Structural Assumption in Input Space.** Another class of methods assumes a low-rank structure in the input space. Many of them introduce *Low-dimensional Embeddings*. Wang et al. (2016) proposed a method named REMBO, building the GP surrogate in a low-dimensional embedding space, $\mathcal{Z} \subset \mathbb{R}^{d_{\text{emb}}}$ where $d_{\text{emb}} \ll d$. The acquisition function is optimized in the embedding space. When querying the function value, the input is recovered by $\mathbf{x} = \mathbf{A}\mathbf{z}$ where $\mathbf{z} \in \mathcal{Z}$, and $\mathbf{A}$ is a random matrix. If the recovered $\mathbf{x}$ is out of the domain $\mathcal{X}$, it will be clipped as a boundary point. To avoid clipping, Nayebi et al. (2019) proposed HESBO, which randomly duplicates $\mathbf{z}$ (and/or flip the signs) to form $\mathbf{x}$. This way, the optimization of $\mathbf{z}$ can respect the same bounding constraints as in $\mathcal{X}$. This essentially imposes further constraints on the embedding space. A more flexible solution, ALEBO, was proposed in (Letham et al., 2020). ALEBO also uses a random matrix $\mathbf{A}$ to recover the

input $\mathbf{x}$. But when maximizing the acquisition function, ALEBO incorporates a constraint that $\mathbf{Az}$ must be in $\mathcal{X}$, thereby avoiding the clipping issue. Another direction is to impose *Partial Variable Dependency* by triming down the input variables. The recent start-of-the-art, SaasBO (Eriksson & Jankowiak, 2021), uses a horse-shoe prior (Carvalho et al., 2009) over the length-scale parameter for each variable. As the horse-shoe prior induces strong sparsity, a massive number of variables can be pruned, substantially reducing the input dimension.

## 4 Theoretical Analysis

To understand why standard BO can fail in high-dimensional cases, in particular with the SE kernel (see Section 6), we look into the gradient structure during the GP training. Specifically, we consider at the beginning, all the length-scale parameters are set to the same initial value, $\ell_1 = \ldots = \ell_d = \ell_0$. From the marginal likelihood (3), we derive the gradient w.r.t each length-scale,

$$\frac{\partial \log p(\mathbf{y}|\mathbf{X})}{\partial \ell_k} = \frac{1}{2}\text{tr}\left(\mathbf{A} \cdot \frac{\partial \mathbf{K}}{\partial \ell_k}\right), \tag{5}$$

where $\mathbf{A} = \boldsymbol{\alpha}\boldsymbol{\alpha}^\top - (\mathbf{K} + \sigma^2\mathbf{I})^{-1}$, and $\boldsymbol{\alpha} = (\mathbf{K} + \sigma^2\mathbf{I})^{-1}(\mathbf{y} - \mathbf{m})$. When using the SE kernel as specified in (1), each entry of $\frac{\partial \mathbf{K}}{\partial \ell_k}$ is as follows: $\left[\frac{\partial \mathbf{K}}{\partial \ell_k}\right]_{ii} = 0$ and for $i \neq j$,

$$\left[\frac{\partial \mathbf{K}}{\partial \ell_k}\right]_{ij} = \frac{\partial \kappa_{\text{SE}}(\mathbf{x}_i, \mathbf{x}_j)}{\partial \ell_k} = \frac{2a}{\ell_k}\exp(-\frac{\|\mathbf{x}_i - \mathbf{x}_j\|^2}{\ell_k^2})\frac{(x_{ik} - x_{jk})^2}{\ell_k^2} \leq \frac{2a}{\ell_k} \cdot \frac{\rho^2}{e^{\rho^2}}, \tag{6}$$

where $x_{ik}$ and $x_{jk}$ are the $k$-th element in $\mathbf{x}_i$ and $\mathbf{x}_j$, respectively, and $\rho = \frac{\|\mathbf{x}_i - \mathbf{x}_j\|}{\ell_0}$.

Intuitively, as the input dimension $d$ increases, the squared distance between input vectors grows rapidly. Consequently, the factor $\frac{\rho^2}{e^{\rho^2}}$ in (6) can quickly fall below the machine epsilon $\xi$, causing the gradient vanishing issue (Bengio et al., 1994; Hanin, 2018). That is, the length-scale cannot be effectively updated according to $\partial \kappa_{\text{SE}}(\mathbf{x}_i, \mathbf{x}_j)/\partial \ell_k$, due to the limitations of the numerical precision.

**Proposition 4.1.** *Given any $\xi > 0$, $\frac{\rho^2}{e^{\rho^2}} < \xi$ when $\rho > \tau_{SE} = \frac{1}{2} + \sqrt{\frac{1}{4} - \log \xi}$.*

When using the double-precision floating-point format (float64), the rounding machine epsilon is $\xi = 2^{-53}$, and the above threshold is $\tau_{\text{SE}} = 6.58$. To analyze the likelihood of gradient vanishing, we provide a probabilistic bound under uniform data distribution.

**Lemma 4.2.** *Suppose the input domain is $[0, 1]^d$ and the input vectors are sampled independently from the uniform distribution, then for any constant threshold $\tau > 0$, when $d > 6\ell_0^2\tau^2$,*

$$p(\rho \geq \tau) > 1 - 2\exp\left(-\frac{(d - 6\ell_0^2\tau^2)^2}{18d}\right). \tag{7}$$

The lower bound grows exponentially with the input dimension $d$ and converges to one. This implies that given any fixed choice of $\ell_0$ relevant to $d$ (e.g., $\ell_0 = 1$), as $d$ increases, the probability $p(\rho \geq \tau_{\text{SE}})$ will rapidly approach one, leading to vanishing gradient for every $\kappa_{\text{SE}}(\mathbf{x}_i, \mathbf{x}_j)$. As a consequence, each length scale $\ell_k$ in (5) cannot be effectively updated, preventing meaningful training[1]. The inferior prediction accuracy of the GP model further contributes to poor performance of the BO procedure. Since $\tau_{\text{SE}}$ is small, gradient vanishing can occur before $d$ becomes extremely large. For example, for $\ell_0 = 0.5$ and $\ell_0 = 1$, the probability $p(\rho > \tau_{\text{SE}})$ exceeds 0.99 when $d \geq 205$ and $d \geq 473$, respectively.

**Why Matérn kernels perform much better?** In our practical evaluations (see Section 6), standard BO with Matérn kernels (2) often perform much better than with the SE kernel in high-dimensional problems, frequently achieving state-of-the-art results. To understand the reasons behind this improvement, we apply the same framework to analyze the behavior of GP training with the Matérn

---

[1]A union bound can be directly derived from (7) to lower bound the joint probability that all gradients $\{\partial \kappa_{\text{SE}}(\mathbf{x}_i, \mathbf{x}_j)/\partial \ell_k\}_{i,j}$ across $N$ training instances are below machine epsilon. However, this bound is too loose to reflect our practical observation (see Fig. 1 and 2 and Table 2). Therefore, we focus our discussion on a single pair of inputs.

| Lower bound | 0.95 | 0.99 | 0.995 | 0.999 | 0.9995 | 0.9999 |
|---|---|---|---|---|---|---|
| SE ($d$) | 172 | 205 | 219 | 250 | 264 | 294 |
| Matérn ($d$) | 980 | 1040 | 1064 | 1116 | 1137 | 1185 |

Table 1: The input dimension $d$ *vs.* lower bound probability of gradient vanishing under $\ell_0 = 0.5$.

kernel. First, we obtain the gradient,

$$\frac{\partial \kappa_{\text{Matérn}}(\mathbf{x}_i, \mathbf{x}_j)}{\partial \ell_k} = \frac{5}{3}(1 + \sqrt{5}\rho)\exp\left(-\sqrt{5}\rho\right)\frac{(x_{ik} - x_{jk})^2}{\ell_k^3} \leq \frac{5(1 + \sqrt{5})\rho}{3\ell_k^3} \cdot \frac{\rho^2}{e^{\sqrt{5}\rho}}.$$

As the distance increases, the factor $\rho^2/e^{\sqrt{5}\rho}$ will also converge to zero, implying that with the growth of $d$, the gradient will eventually fall below machine epsilon. However, since this factor $\rho^2/e^{\sqrt{5}\rho}$ decreases much slower than the corresponding factor to the SE kernel, namely $\rho^2/e^{\rho^2}$ in (6), the Matérn kernel is able to robustly handle much higher dimensions.

**Proposition 4.3.** *Given $\forall \xi > 0$, $\frac{\rho^2}{e^{\sqrt{5}\rho}} < \xi$ when $\rho > \tau_{Matérn} = \left(1 + \sqrt{1 + \log 1/5 - \log \xi}\right)^2 / \sqrt{5}$.*

With the machine epsilon $\xi = 2^{-53}$, the threshold $\tau_{\text{Matérn}} = 21.98$, more than $300\%$ larger than $\tau_{\text{SE}} = 6.58$. Therefore, to achieve the same probability bound in (7), a much higher $d$ is required for Matérn kernel compared to the SE kernel under the same $\ell_0$. For concrete examples, we list the dimension $d$ versus the probability bound in Table 1. It implies that using the same length-scale initialization, the Matérn kernel is able to robustly handle higher dimensions.

**Numerical Verification.** To verify whether gradient vanishing occurs during practical GP training and to access whether our analysis aligns with the empirical observations, we conducted numerical experiments using two synthetic functions, Hartmann6($d$, 6) and Rosenbrock(d,d) (see Section 6.1 for definitions). We varied the input dimension $d$ across {50, 100, 200, 300, 400, 500, 600}, with the domain set to $[0, 1]^d$. For each value of $d$, we uniformly sampled 500 training inputs and 100 test inputs. We trained GP with both SE and Matérn kernels, using length-scale initialization $\ell_0$ from $\{0.1, 0.5, 0.693, 1.0, \sqrt{d}\}$. Note $\ell_0 = 0.693 = \text{SoftPlus}(0)$ is a popular initialization choice, used as the default choice in influential GP/BO libraries GPyTorch (Gardner et al., 2018) and BOTorch (Balandat et al., 2020). Training and testing were repeated 20 times for each $d$ and $\ell_0$. We then evaluated **(A)** the average relative $L_2$ difference between the length-scale vector before and after training, namely, $\|\ell_{\text{trained}} - \ell_{\text{init}}\| / \|\ell_{\text{init}}\|$, **(B)** the average gradient norm at the first training step, and **(C)** the average test Mean-Squared-Error (MSE).

As shown in Fig. 1, gradient vanishing indeed occurred, leading to training failure. When using $\ell_0 = 0.1$ for GP training with the SE and Matérn kernels, both the length-scale relative $L_2$ difference and the gradient norm remained close to zero across all choices of $d$ (those values are not displayed because they are too small). Consequently, the MSE hovered around 1.0, indicating consistent training failure. When we increased $\ell_0$ to 0.5 and 1.0 for training with the SE kernel, both the length-scale difference and gradient norm became significantly larger and non-trivial for $d = 50$ and $d = 100$, resulting in MSE below 0.2. This demonstrates that gradient vanishing did not occur and training was successful. However, as $d$ increased, gradient vanishing reappeared at $d = 200$ for $\ell_0 = 0.5$ and at $d = 400$ for $\ell_0 = 1.0$, for both Hartmann6 and Rosenbrock. In these cases, the length-scale difference and gradient norm dropped near zero, while the MSE returned to around 1.0. A similar trend was observed with the Matérn kernel. However, GP training with the Matérn kernel can succeed with higher dimensions under the same initialization conditions. For example, with $\ell_0 = 0.5$, for both Hartmann6 and Rosenbrock, training with the Matérn kernel started to failed until $d = 400$, compared to $d = 200$ with the SE kernel. When using $\ell_0 = 1.0$, the Matérn kernel succeeded across all $d$ values, while the SE kernel started to fail at $d = 400$. Table 2 summarizes the input dimensions at which gradient vanishing and training failure began to happen.

**Confirmation in BO Running.** We next investigated whether gradient vanishing occurs during the BO process, potentially leading to poor optimization performance. To this end, we examined two benchmark BO tasks: SVM (D=388) and Rosenbrock (300,100), as detailed in Section 6. As shown in Fig. 3 and 4, standard BO with the Matérn kernel (SBO-Matérn) achieved top-tier results, while BO with the SE kernel (SBO-SE) performed poorly. Both methods used the initialization $\ell_0 = 0.693$. We then extracted the BO trajectories for both methods and analyzed the relative $L_2$ difference in the length-scale vectors before and after training, and the gradient norm at the first training step.

As illustrated in Fig. 2, during the BO iterations, GP-SE ($\ell_0 = 0.693$) consistently experienced gradient vanishing and so training failure, whereas GP-Matérn ($\ell_0 = 0.693$) maintained non-trivial gradients. This difference directly correlates with their respective optimization performances — poor for SBO-SE and excellent for SBO-Matérn.

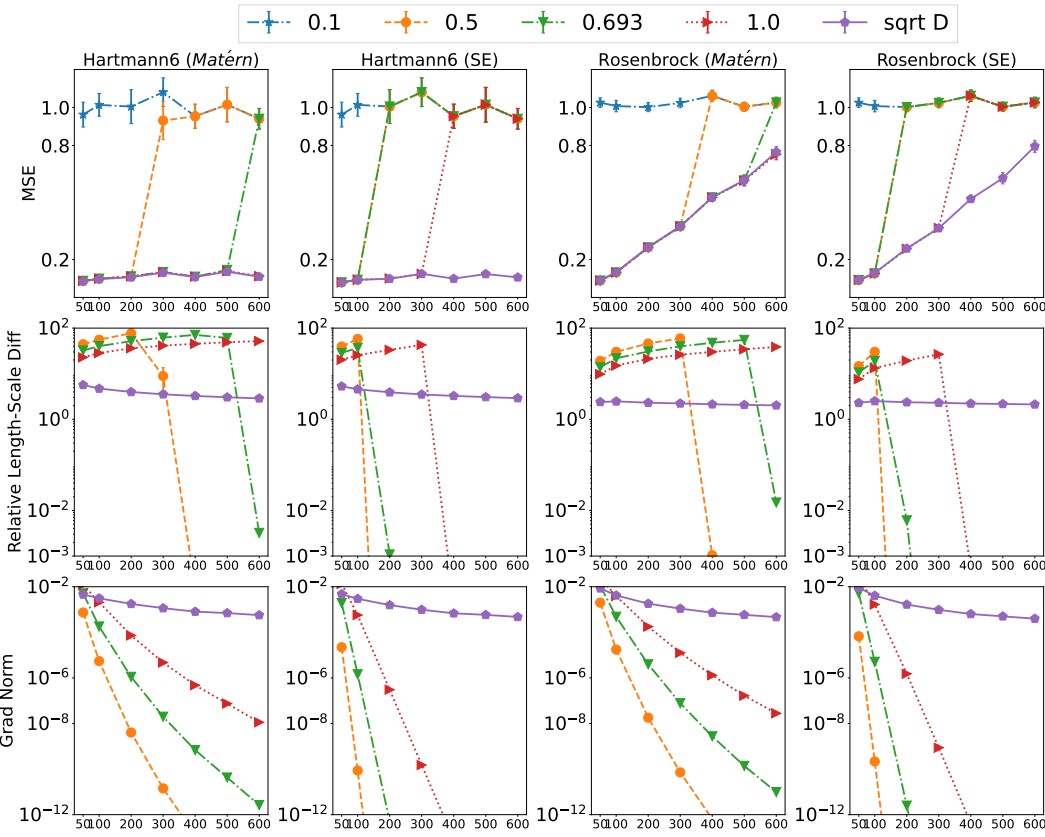

Figure 1: Mean Square Error (MSE), Relative $L_2$ difference between the length-scale vectors before and after training, and $L_2$ norm of the length-scale gradient at the first training step, across different input dimensions ($x$-axis) and initializations (legend). The results were averaged from 20 runs. For $\ell_0 = 0.1$, the length-scale relative difference and gradient norm are not displayed, as they fall much below the minimum values on the $y$-axis.

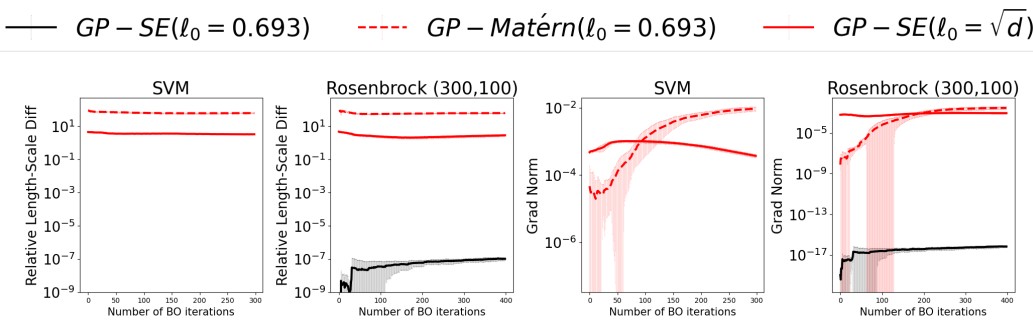

Figure 2: The relative $L_2$ difference of the length-scale vectors before and after training and the gradient norm at the first training iteration across every step of BO. The results for GP-SE ($\ell_0 = 0.693$) are not displayed, as they fall much below the minimum values on the y-axis.

## 5 Robust Initialization

From the analysis in Section 4, one straightforward approach to improving learning with the SE (and also Matérn) kernel is to initialize each length-scale with a larger $\ell_0$, which can loosen the lower

| Hartmann6 | $\ell_0 = 0.1$ | 0.5 | 0.6931 | 1.0 | $\sqrt{d}$ |
|---|---|---|---|---|---|
| SE | 50 | 200 | 200 | 400 | ✓ |
| Matérn | 50 | 400 | 600 | ✓ | ✓ |
| Rosenbrock | $\ell_0 = 0.1$ | 0.5 | 0.6931 | 1.0 | $\sqrt{d}$ |
| SE | 50 | 200 | 200 | 400 | ✓ |
| Matérn | 50 | 400 | 600 | ✓ | ✓ |

Table 2: The input dimension from which gradient vanishing started to occur and training started to fail; ✓ means no gradient vanishing occurred.

bound in (7) and allow the GP model to better adapt to higher dimensions. However, this approach is not robust enough, as $d - 6\ell_0^2\tau^2$ will continue to increase with $d$, and the bound will still rise toward one rapidly, causing the gradient to vanish. To address this issue, we propose setting $\ell_0$ to growing with $d$ rather than keeping it constant,

$$\ell_0 = c\sqrt{d}, \quad c > 0. \tag{8}$$

**Lemma 5.1.** *Suppose the input domain is $[0,1]^d$ and each input vector is independently sampled from the uniform distribution. Given any constant threshold $\tau > 0$, we set $\ell_0 = c\sqrt{d}$ such that $c > \frac{1}{\sqrt{6}\tau}$, then*

$$p(\rho \geq \tau) \leq 2\exp\left(-2(c^2\tau^2 - \frac{1}{6})^2 d\right) \propto \exp(-\mathcal{O}(d)). \tag{9}$$

With this initialization, the increase in $d$ *exponentially* reduces the upper bound on the probability of $\rho$ exceeding the given threshold $\tau$. As a result, the gradient vanishing issue can be fundamentally mitigated in the high-dimensional settings.

In our experiments, we tested standard BO using SE kernel with our proposed length-scale initialization as specified in (8) where we set $c = 1$. As shown in Fig. 3 and Fig. 4, the performance across all the benchmarks is dramatically improved, as compared to using popular initialization $\ell_0 = 0.693$. The performance matches that of using Matérn kernels, achieving state-of-the-art (or near-best) results. In addition, as shown in Fig. 1 and Table 2, our numerical experiments confirm that gradient vanishing never occurred with our robust initialization method. Fig. 2 further demonstrates that our method eliminates gradient vanishing during practical BO runs, enabling effective GP training and much better optimization results. Together this shows that our method can effectively adapt SE kernels to high-dimensional settings, and mitigate the failure mode of standard BO.

**Alternative method.** Recent work (Hvarfner et al., 2024) proposes an intuitive approach to address the learning challenge of the SE kernel in the high-dimensional spaces by constructing a log-normal prior over each length scale, $p(\ell_k) = \text{LogNormal}(\ell_k | \mu_0 + \frac{\log(d)}{2}, \sigma_0)$. This prior regularizes the length-scale to be on the order of $\sqrt{d}$ during training. Using our theoretical framework, this method can be justified as an alternative strategy to alleviate the gradient vanishing issue. However, our method, directly motivated by our analysis, is even simpler — requiring no additional prior construction or associated parameters, and has also shown effective in addressing the gradient vanishing issue. The comparison with (Hvarfner et al., 2024) is given in Section 6.

## 6 Comprehensive Evaluation

### 6.1 Experimental Settings

For a comprehensive evaluation of the standard BO, we employed twelve benchmarks, of which six are synthetic and six are real-world problems. For clarity, all the tasks aim for function maximization.

**Synthetic Benchmarks.** We considered four popular synthetic functions: Ackley, Rosenbrock, Hartmann6, and Stybtang. Definitions of these functions are provided in Section B.1 of the Appendix. Each task is represented by "Fun$(d, d')$", where $d$ is the input dimension that BO methods optimize for, and $d'$ is the number of effective variables ($d' \leq d$). The ground-truth of the target function

values is computed with the first $d'$ variables. The benchmarks and the structures within the target functions are summarized in Appendix Table 3.

**Real-World Benchmarks.** We employed the following real-world benchmark problems: **Mopta08 (124)** (Jones, 2008), a vehicle design problem. The objective is to minimize the mass of a vehicle with respect to 124 design variables, subject to 68 performance constraints. We followed (Eriksson & Jankowiak, 2021) to encode the constraints as a soft penalty, which is added into the objective. **SVM (388)** (Eriksson & Jankowiak, 2021), a hyper-parameter tuning problem that optimizes 3 regularization parameters and 385 length-scale parameters for a kernel support vector machine regression model. **Rover (60)**, a rover trajectory planning problem from (Wang et al., 2018). The goal is to find an optimal trajectory determined by the location of 30 waypoints in a 2D environment. **DNA (180)** (vSehić et al., 2022), a hyper-parameter optimization problem that optimizes 180 regularization parameters for weighted lasso regression on an DNA dataset (Mills, 2020). Prior analysis (vSehić et al., 2022) shows that only around 43 variables are relevant to the target function. **NAS201 (30)** (Dong & Yang, 2020), a neural architecture search problem on CIFAR-100 dataset. **Humanoid Standup (1003)**: A novel trajectory optimization benchmark based on a humanoid simulator that uses the MuJoCo physics engine (Todorov et al., 2012). The problem dimension is 1,003. The details are provided in Appendix Section B.2.

**Methods.** We tested with four versions of standard BO:

- *SBO-Matérn*: GP regression with ARD Matérn-5/2 kernel as specified in (2), the length-scale initialization $\ell_0 = \mathrm{SoftPlus}(0) = 0.693$, as the default choice in the influential GP/BO libraries GPyTorch and BOTorch.
- *SBO-SE*: GP regression with ARD SE kernel as specified in (1), the length-scale initialization $\ell_0 = \mathrm{SoftPlus}(0) = 0.693$.
- *SBO-Matérn (RI)*: GP regression with ARD Matérn-5/2 kernel, using our proposed robust initialization $\ell_0 = c\sqrt{d}$ where $d$ is the input dimension and $c = 1$.
- *SBO-SE (RI)*: GP regression with ARD SE kernel, using our robust initialization with $c = 1$.

We used UCB as the acquisition function where the exploration level $\lambda$ was set to 1.5. We used GPyTorch for GP training and BOTorch for Bayesian optimization. To ensure efficiency, the GP was trained via point estimation of the length-scale and noise variance parameters, with the optimizer selected from L-BFGS, Adam, or RMSProp. Further details are provided in Appendix Section B.3.

We compared with six state-of-the-art high-dimensional BO methods, including *Tree UCB* (Han et al., 2021), *RDUCB* (Ziomek & Ammar, 2023), *HESBO* (Nayebi et al., 2019), *ALEBO* (Letham et al., 2020), *SaasBO* (Eriksson & Jankowiak, 2021), and *TURBO* (Eriksson et al., 2019). The first five have been introduced in Section 3. *TURBO* is designed for a special scenario where the target function can be extensively evaluated, *e.g.,* tens of thousands evaluations. This is uncommon in BO applications, as the function evaluation is typically deemed expensive, and one aims to evaluate the target function as few as possible. *TURBO* searches for trust-regions in the input domain, learns a set of local GP models over each region, and then uses an implicit bandit strategy to decide which local optimization runs to continue. In addition, we tested the most recent method *VBO* (*Vanilla BO*) (Hvarfner et al., 2024) that levels up the performance of the standard BO by constructing a log-normal prior over the length-scale parameters (see Section 5). We used the original implementation and default (recommended) settings for the competing methods. Besides, we compared with two BO methods based on Bayesian neural networks (BNN). One uses Laplace approximation and the other Hamiltonian Monte-Carlo (HMC) sampling for posterior inference. We used a high-quality open-source implementation for each BNN based method. Appendix Table 4 summarizes the structural assumptions made by all the methods. The details of all the competing methods are given in Appendix Section B.3.

For each optimization task, we randomly queried the target function 20 times to collect the initial data, except for Humanoid-Standup, where we collected 50 initial data points. We tested TURBO with a single trust region and five trust regions, denoted as TURBO-1 and TURBO-5 respectively. For HESBO and ALEBO, we varied $d_{\mathrm{emb}} \in \{10, 20\}$ and denote the choice as HESBO/ALEBO-{10, 20}. For SaasBO, we performed tests using both NUTS and MAP estimation for surrogate training, denoted by SaasBO-NUTS and SaasBO-MAP, respectively. For a reliable comparison, we ran each method ten times, ensuring that all methods used the same randomly collected initial dataset in each run (these datasets varied across different runs). Additionally, we conducted ten extra runs for all the standard BO methods and VBO to further validate their performance.

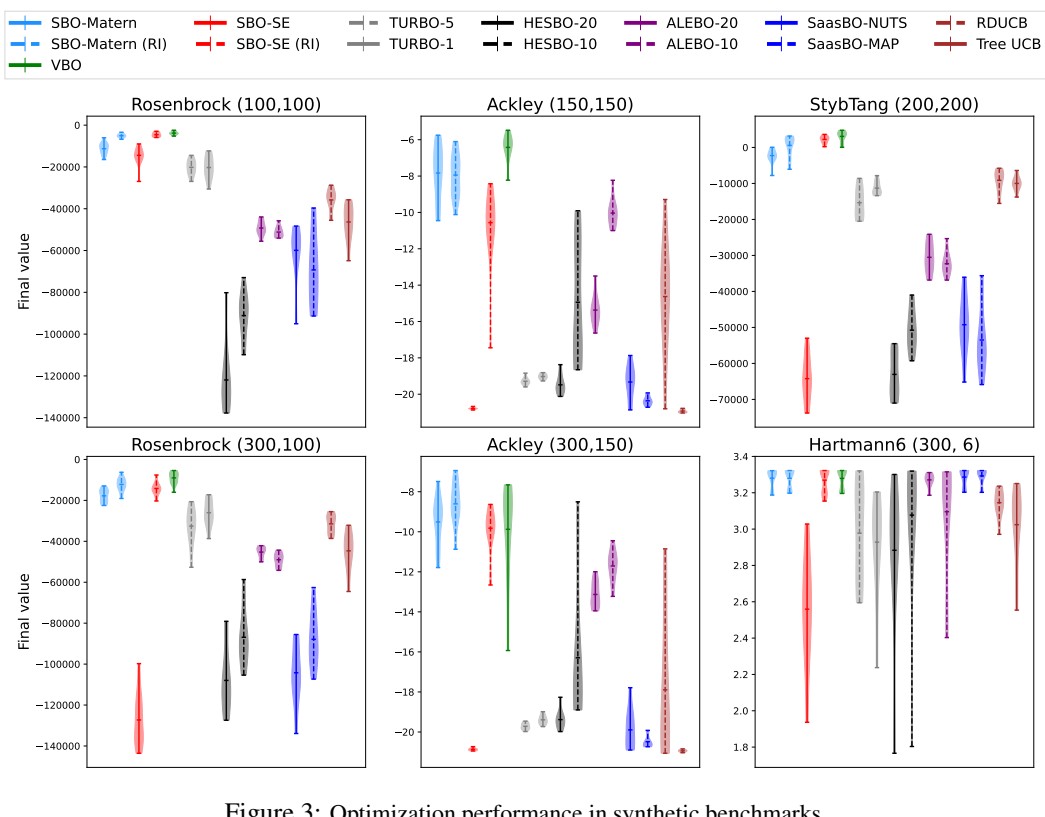

Figure 3: Optimization performance in synthetic benchmarks.

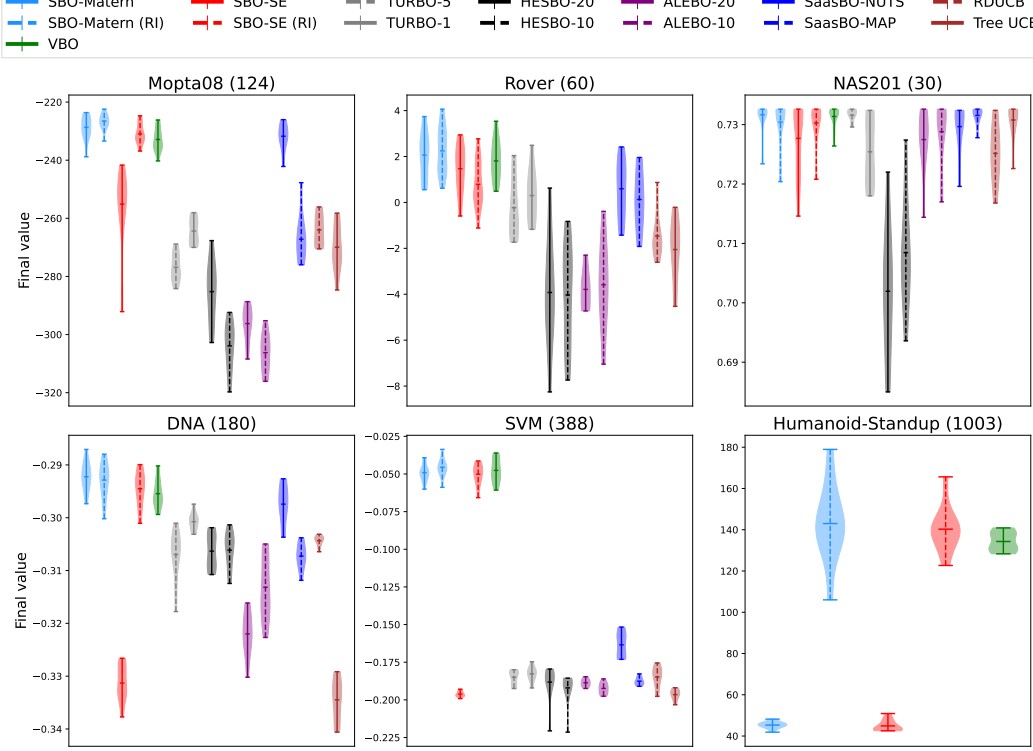

Figure 4: Optimization performance in real-world problems.

### 6.2 Optimization Performance

**Ultimate Outcomes.** We first examined the ultimate optimization outcomes of each method. The results are reported in Fig. 3 for synthetic benchmarks and in Fig. 4 for real-world benchmarks. Due to space limit, we supplement the BNN-based BO results in Appendix Section C.1. We used violin plots that illustrate the average of the obtained maximum function value over the runs (the middle bar), the best (the top bar), the worst (the bottom bar), and the distribution of the obtained maximum (the shaded region).

On average, *SBO-Matérn (RI)*, *SBO-SE (RI)* and *VBO* consistently deliver top-tier results, often outperforming the other methods by a large margin, *e.g.,* on benchmarks Ackley (300, 150), DNA, and SVM. It is noteworthy that SBO-Matérn, despite using a small length-scale initialization ($\ell_0 = 0.693$), also achieved top-tier performance across all benchmarks except for Humanoid-Standup, where the optimization dimension exceeds 1,000. In contrast, *SBO-SE* failed when the input dimension $d \geq 150$ due to the gradient vanishing issue, *e.g.,* Ackley(150, 150), Rossenbrock(300, 100), DNA, and SVM (see our theoretical analysis and numerical experiments in Section 4). The result verifies that the Matérn kernel is far less susceptible to gradient vanishing and performs much more robustly for high-dimensional problems than the SE kernel. However, when the dimension is extremely high (*e.g.,* over 1K), the small length-scale initialization can still cause vanishing gradients and degraded optimization performance. After applying our robust initialization, both kernels achieved top-tier performance consistently, closely matching VBO.

**Runtime Performance.** Next, we examined the runtime performance. Appendix Fig. 5 and 6 report the average maximum function value obtained at each step and the standard deviation across all runs. Since we have 16 methods for comparison, showing the standard deviation for every method makes the figure cluttered. To ensure clarify, we show the curves and standard deviation of SBO, VBO, and the best, worst, and median performed remaining baselines so far at each step.

After an initial stage, *SBO-Matérn (RI)*, *SBO-SE (RI)*, and *VBO* consistently produced superior queries, achieving larger function values. So did *SBO-Matérn* in all the cases except Humanoid-Standup. Their performance curves are generally above those of competing methods, reflecting a more efficient optimization of the target function. This trend is particularly noticeable in benchmarks such as Ackley (300, 150), Rosenbrock (300, 100), and SVM. The performance of SBO-SE degraded when $d \geq 150$, with its curve positioned toward the bottom due to the gradient vanishing issue. In benchmarks such as Rosenbrock (100, 100) and DNA (180), the curve for *VBO* is predominantly above those of *SBO-Matérn (RI)* and *SBO-SE (RI)*, indicating even more efficient optimization. However, on Ackley (150, 150), Ackley (300, 150) and Humanoid-Standup (1003), *VBO*'s curve is mostly below that of *SBO-Matérn (RI)* and *SBO-SE (RI)*, and exhibits worse performance. For the remaining benchmarks, their curves are close and often overlap. Overall, in all cases, they tend to converge to similar function values.

These results collectively demonstrate that standard BO can excel in high-dimensional optimization problems. The lack of additional structural assumptions in GP modeling may make it more flexible in capturing various correlations within high-dimensional spaces. The primary challenge for standard BO arises from the gradient vanishing issue, which is often due to improper initialization of the length-scales. By using our robust initialization method, this risk can be significantly mitigated.

**Additional Results.** We conducted extensive additional evaluations to examine exploration parameters in UCB, and other acquisition functions on SBO performance in high-dimensional settings. We found that UCB is notably robust to variations in the exploration parameter. Additionally, log-EI facilitates strong performance in high-dimensional optimization, whereas EI and Thompson Sampling (TS) often lead to subpar results. Detailed results and discussions are provided in Appendix Section C.

## 7 Conclusion

We conducted a thorough investigation of standard BO in high-dimensional optimization, both empirically and theoretically. Our analysis identified gradient vanishing as a major failure mode of standard BO, and we proposed a simple yet robust initialization method to address this issue. Our empirical evaluation demonstrates that, once this problem is mitigated, standard BO can achieve state-of-the-art performance in high-dimensional optimization.

## Acknowledgement

This work has been supported by NSF CAREER Award IIS-2046295, NSF OAC-2311685, and Margolis Foundation.

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

# A Proofs

## A.1 Proof of Proposition 4.1

*Proof.* We intend to solve $\rho^2/e^{\rho^2} < \xi$, which is equivalent to solving

$$\log r - r < \log \xi,$$

where $r = \rho^2$. We then leverage the fact that $\log r < \sqrt{r}$ when $r > 1$. Since $1/e > \xi$, we know the solution of $\rho$ must be larger than 1, implying that the solution for $r$ is larger than 1. To meet the inequality, we can solve

$$\log r - r < \sqrt{r} - r < \log \xi.$$

It then converts to a quadratic form,

$$\sqrt{r} - \sqrt{r}^2 < \log \xi. \tag{10}$$

Note that $\rho = \sqrt{r}$. It is straightforward to solve this inequality and we can obtain that the inequality holds when

$$\rho > \tau_{\text{SE}} = \frac{1}{2} + \sqrt{\frac{1}{4} - \log \xi}.$$

$\square$

## A.2 Proof of Lemma 4.2

*Proof.* First, since each $x_{ik}$ and $x_{jk}$ independently follow Uniform$(0, 1)$, it is straightforward to obtain $\mathbb{E}[(x_{ik} - x_{jk})^2] = \frac{1}{6}$. Let us define $\gamma = \|\mathbf{x}_i - \mathbf{x}_j\|^2$. Then we have $\mathbb{E}[\gamma] = \frac{1}{6}d$. According to Hoeffding's inequality, for all $t > 0$,

$$p(|\gamma - \frac{1}{6}d| \geq t) \leq 2 \exp\left(-\frac{2t^2}{d}\right). \tag{11}$$

Since $\rho = \frac{\sqrt{\gamma}}{\ell_0}$, we have $p(\rho > \tau) = p(\gamma > \tau^2\ell_0^2)$. Since $d > 6\ell_0^2\tau^2$, we set $t = \frac{1}{6}d - \ell_0^2\tau^2 > 0$, and apply (11) to obtain

$$p(|\gamma - \frac{1}{6}d| \geq \frac{1}{6}d - \ell_0^2\tau^2|) \leq 2 \exp\left(-\frac{2\left(\frac{1}{6}d - \ell_0^2\tau^2\right)^2}{d}\right) = 2 \exp\left(-\frac{\left(d - 6\ell_0^2\tau^2\right)^2}{18d}\right),$$

which is equivalent to

$$p(|\gamma - \frac{1}{6}d| < \frac{1}{6}d - \ell_0^2\tau^2) > 1 - 2 \exp\left(-\frac{\left(d - 6\ell_0^2\tau^2\right)^2}{18d}\right). \tag{12}$$

Since $p(\gamma > \ell_0^2\tau^2) \geq p(\ell_0^2\tau^2 < \gamma < \frac{1}{3}d - \ell_0^2\tau^2) = p(|\gamma - \frac{1}{6}d| < \frac{1}{6}d - \ell_0^2\tau^2)$, combining with (12) we obtain the bound

$$p(\rho > \tau) = p(\gamma > \ell_0^2\tau^2) > 1 - 2 \exp\left(-\frac{\left(d - 6\ell_0^2\tau^2\right)^2}{18d}\right).$$

$\square$

## A.3 Proof of Proposition 4.3

*Proof.* To solve $\rho^2/e^{\sqrt{5}\rho} < \xi$, it is equivalent to solving

$$\frac{1}{5}\frac{r^2}{e^r} < \xi, \quad r = \sqrt{5}\rho,$$

which is further equivalent to

$$\log \frac{1}{5} + 2\log r - r < \log \xi.$$

| Fun$(d, d')$ | Structure |
|---|---|
| **Ackley(300, 150)** | Partial variable dependency |
| **Rosenbrok(300, 100)** | Partial variable dependency
Nonoverlap additive decomposition |
| **Hartmann6(300,6)** | Partial variable dependency |
| **Rosenbrock(100, 100)**
**Stybtang(200, 200)** | Nonoverlap additive decomposition |
| **Ackley(150, 150)** | None |

Table 3: Synthetic benchmarks and the structures within the target function: $d$ is the input dimension, and $d'$ is the number of effective variables to compute the target function.

We leverage $\log r < \sqrt{r}$ for $r > 1$, and solve the upper bound,

$$\log \frac{1}{5} + 2\sqrt{r} - r < \log \xi,$$

which gives

$$\rho > \tau_{\text{Matérn}} = \frac{1}{\sqrt{5}} \left( 1 + \sqrt{1 + \log 1/5 - \log \xi} \right)^2. \tag{13}$$

$\square$

### A.4 Proof of Lemma 5.1

*Proof.* First, similar to the proof in Section A.2, under the uniform distribution on $[0,1]^d$, from Hoeffding's inequality, we have for all $t > 0$,

$$p(|\gamma - \frac{1}{6}d| \geq t) \leq 2 \exp \left( -\frac{2t^2}{d} \right). \tag{14}$$

When we set $\ell_0 = c\sqrt{d}$ such that $c > 1/\sqrt{6}\tau$, we have $\ell_0^2\tau^2 > d/6$. Let us choose $t = \ell_0^2\tau^2 - d/6 = c^2\tau^2 d - d/6 > 0$, and apply the Hoeffding's inequality (14),

$$p \left( |\gamma - d/6| \geq \ell_0^2\tau^2 - d/6 \right) \leq 2 \exp \left( -\frac{2(\ell_0^2\tau^2 - d/6)^2}{d} \right) = 2 \exp \left( -2(c^2\tau^2 - \frac{1}{6})^2 d \right). \tag{15}$$

Since $p(\rho \geq \tau) = p(\gamma \geq \ell_0^2\tau^2) \leq p(|\gamma - d/6| \geq \ell_0^2\tau^2 - d/6)$, we obtain

$$p(\rho \geq \tau) \leq 2 \exp \left( -2(c^2\tau^2 - \frac{1}{6})^2 d \right).$$

$\square$

## B  More Experiment Details

### B.1  Definition of Synthetic Functions

**Stybtang**. We used the following slightly-modified Stybtang function,

$$f(\mathbf{x}) = \frac{1}{2} \sum_{i=1}^{D} \left( (x_i - c_i)^4 - 16(x_i - c_i)^2 + 5(x_i - c_i) \right),$$

where $\mathbf{x} \in [-5, 5]^D$, and we set $[c_1, \ldots, c_D]$ as an evenly spaced sequence in $[0, 7.5]$ ($c_1 = 0$ and $c_D = 7.5$). The optimum is at $\mathbf{x}^\dagger = [c_1 - 2.903534, \ldots, c_D - 2.903534]$.

**Rosenbrock**. We used the following Rosenbrock function,

$$f(\mathbf{x}) = \sum_{i=1}^{D-1} \left[ 100 \left( (x_{i+1} - c_{i+1}) - (x_i - c_i)^2 \right)^2 + (1 - (x_i - c_i))^2 \right],$$

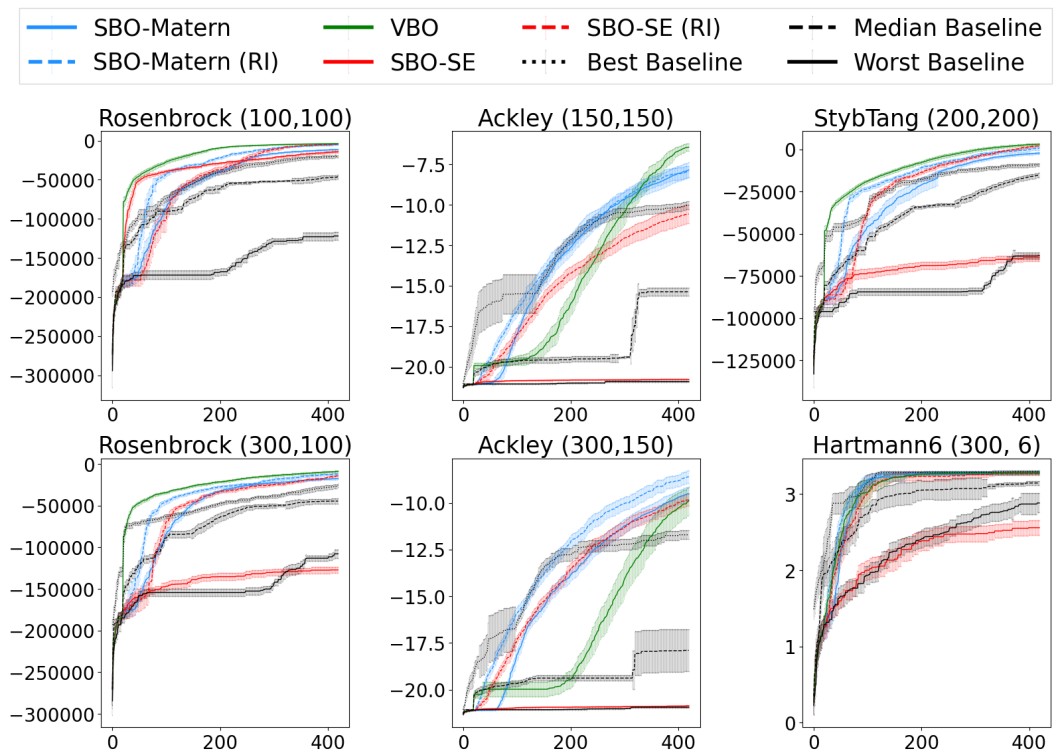

Figure 5: Runtime performance: Maximum Function Value Obtained *vs.* Number of Steps in all synthetic benchmarks. For cleaner view, we plotted the full results only for SBO and VBO. For the remaining methods, we show best, median, and worst result at each step.

where $\mathbf{x} \in [-2.048, 2.048]^D$. We set $[c_1, ..., c_D]$ as an evenly spaced sequence in $[-2, 2]$, where $c_1 = -2$ and $c_D = 2$. The optimum is at $x^\dagger = [c_1 + 1, ..., c_D + 1]$.

**Ackley**. We used the Ackley function defined in (Surjanovic & Bingham),

$$f(\mathbf{x}) = -20 \exp\left(-0.2\sqrt{\frac{1}{d}\sum_{i=1}^{d} x_i^2}\right) - \exp\left(\frac{1}{d}\sum_{i=1}^{d} \cos(2\pi x_i)\right) + 20 + \exp(1),$$

where each $x_i \in [-32.768, 32.768]$ and the (global) optimum is at $x^\dagger = [0, \ldots, 0]$.

**Hartmann6**. The function is given by

$$f(\mathbf{x}) = -\sum_{i=1}^{4} \alpha_i \exp\left(-\sum_{j=1}^{6} A_{ij}(x_j - P_{ij})^2\right),$$

where each $x_i \in [0, 1]$, $\mathbf{A} = [A_{ij}]$ and $\mathbf{P} = [P_{ij}]$ are two pre-defined matrices as in (Surjanovic & Bingham). The global optimum is at $\mathbf{x}^\dagger = [0.20169, 0.150011, 0.476874, 0.275332, 0.6573]$.

## B.2 Humanoid-Standup Benchmark

We created a novel trajectory optimization benchmark in a Humanoid Standup task based on the MuJoCo physics engine (Todorov et al., 2012). The action for the environment is determined by 17 parameters (corresponding to 17 motors in the humanoid). We set the trajectory length to 59. Therefore, the dimension of the problem is $59 \times 17 = 1,003$. The goal is to find a trajectory $\tau = (\mathbf{a}_1, \ldots, \mathbf{a}_{59})$ of motor actions that maximize the reward, for a given initial state. This is an instance of optimal control and planning, which is a classical problem in reinforcement learning. It is noteworthy that the recent work of Hvarfner et al. (2024) also created a BO benchmark (named

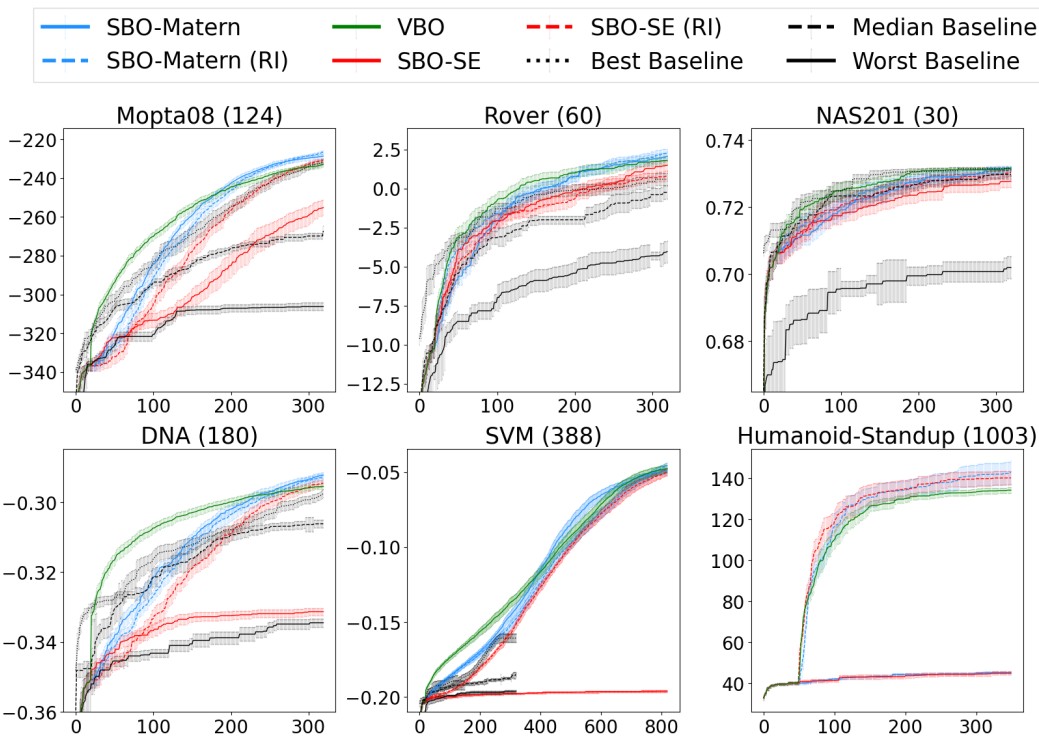

Figure 6: Runtime performance: Maximum Function Value Obtained *vs.* Number of Steps in all real-world benchmarks. For cleaner view, we plotted the full results only for SBO and VBO. For the remaining methods, we show best, median, and worst result at each step.

"Humanoid") based on the MuJoCo engine, but the problem setting is different. Their benchmark seeks to optimize a linear policy, represented by a $376 \times 17$ parameter matrix that linearly maps the humanoid state to an action at each step. In contrast, our benchmark makes no assumptions about an underlying policy and directly optimizes the entire trajectory, making it policy-free.

## B.3 Implementation

- **ALEBO.** We used ALEBO implementation shared by the Adaptive Experimentation (AX) Platform (version 0.2.2). The source code is at `https://github.com/facebook/Ax/blob/main/ax/models/torch/alebo.py`.

- **HESBO.** We used HESBO implementation of the original authors (`https://github.com/aminnayebi/HesBO`).

- **TURBO.** We used the original TURBO implementation (`https://github.com/uber-research/TuRBO`)

- **SaasBO-NUTS.** We used the original implementation of the authors (`https://github.com/martinjankowiak/saasbo`).

- **SaasBO-MAP.** The SaasBO implementation available to the public does not include the version using MAP estimation. We therefore implemented this method based on the original paper (Eriksson & Jankowiak, 2021). All the hyperparameter settings follow exactly the same as the original paper.

- **RDUCB and Tree UCB.** The implementation of both methods is publicly available at `https://github.com/huawei-noah/HEBO/tree/master/RDUCB`.

- **BNN Laplace and BNN HMC.** For BNN Laplace, we used the implementation at `https://github.com/wiseodd/laplace-bayesopt`, and for BNN HMC, we used the implementation at `https://github.com/yucenli/bnn-bo`. To identify the architecture of the neural network, we perform leave-one-out cross-validation on the

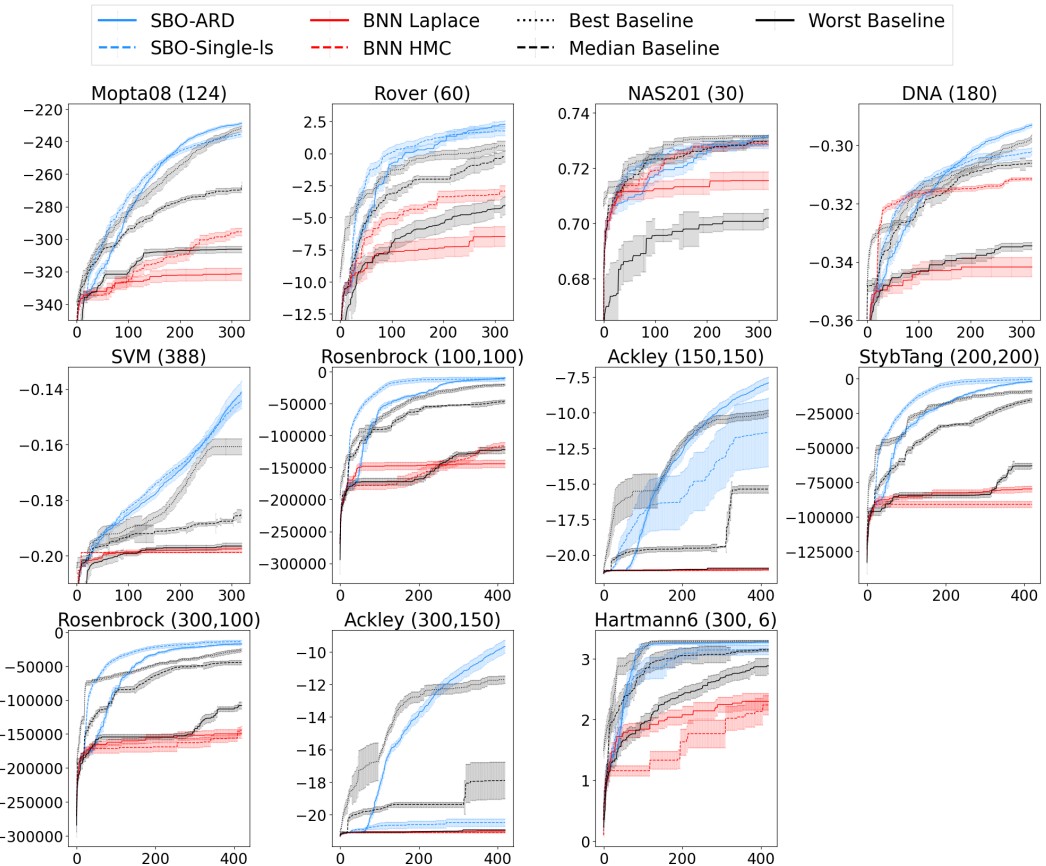

Figure 7: Runtime performance comparison with BNN-based BO methods and SBO with a single length-scale parameter. Matérn kernel was used to conduct SBO. For clarity, VBO was excluded for comparison. The results were averaged from 10 runs. The shaded region depicts the standard deviation.

initial dataset for each task. The layer width and depth were selected from $\{1, 2, 3\} \times \{32, 64, 128, 256\}$. The activation function for BNN Laplace is ReLU and for BNN HMC is Tanh, which is the default choice of each method. For BNN Laplace, the training was conducted by maximizing marginal likelihood with ADAM, with learning rate of 1E-01, and weight decay of 1E-03. For HMC, the BNN is pre-trained by maximizing log likelihood with ADAM. Then followed by HMC sampling procedure. We used their adaptive step size schedule for the Leap frog, and the number of steps was set to 100.

- **GP Training in Standard BO.** For efficiency, we trained GP via point estimation. The optimizer was chosen from L-BFGS, Adam, and RMSProp. For Adam and RMSProp, we set the initial learning rate to 0.1 and 0.01, respectively. The maximum number of epochs was set to 1,500 for both methods. We maximize the log marginal likelihood of the GP regression model. The positiveness of the length-scale and noise variance parameters are ensured via SoftPlus transform. For L-BFGS, we found that maximizing the log marginal likelihood often incurs numerical issues. To achieve numerical stability, we employed the prior `Uniform`(0.001, 30) over each length-scale, a diffused Gamma prior over the noise variance `Gamma`(1.1, 0.05), and the prior `Gamma`(2.0, 0.15) over the amplitude. The maximum number of iterations was set to 15K and tolerance level 2E-9.

## B.4 Computing Environment

We conducted the experimental investigations on a large computer cluster equipped with Intel Cascade Lake Platinum 8268 chips.

| Method | Structural Assumption |
|---|---|
| ALEBO, HESBO | Low-dim embedding |
| SaasBO | Partial variable dependency |
| RDUCB, Tree UCB | Additive function decomposition |
| SBO, TURBO, BNN | None |

Table 4: BO methods and their structural assumption about the target function.

# C  Additional Results

## C.1  BNN-based BO and Standard BO with a Single Length-Scale

Due to the space limit, we did not report the results of the BNN-based BO in the main paper. Here, we supplement the comparison with the aforementioned two BNN-based BO approaches, *BNN Laplace* and *BNN HMC*, as specified in Section B.3. In addition, we also tested SBO with a single length-scale. We used the Matérn kernel. The comparison was performed across all the synthetic and real-world benchmarks except Humanoid-Standup (1003). The results are reported in Fig. 7. Across all the benchmarks, the performance of the BNN based methods is worse than the median of GP based methods. In cases, such as SVM, Hartmann6 (300, 6) and Rosenbrock, the performance of the BNN based methods is close to or overlapping with the performance of the worst GP baselines. This might be due to that under low data regimes, it is more challenging for a neural network to capture the landscape of the target function and quantify the uncertainty, especially for high-dimensional targets.

Interestingly, even using a single length-scale parameter, the standard BO, namely SBO-Single — as long as it does not encounter gradient vanishing — still consistently shows superior performance, which often closely matches ARD Matérn, except for Ackley(300, 150), SBO-Single struggled in finding right locations to query, leading to poor performance.

## C.2  Different Choice of $\lambda$ in UCB

We examined how the exploration parameter $\lambda$ in UCB influences the BO performance in high-dimensional optimization. To this end, we used the Maérn kernel and varied $\lambda$ from {0.5, 1.0, 1.5, 2.0}. We show the runtime optimization performance in Figure 8. It can be seen that, in most benchmarks (Mopta08, Rover, NAS201, DNA, *etc.*), the performance with different $\lambda$'s is close. However, on SVM and Ackley, the performance of $\lambda = 0.5$ is typically worse than the bigger choice of $\lambda$. This might be due to that those problems are more challenging (*e.g.,* Ackley has many local minima), a larger exploration level ($\lambda$) is needed to identify better optimization outcomes.

## C.3  Alternative Acquisition Functions

Finally, we evaluated three other acquisition functions for high-dimensional optimization: Expected Improvement (EI), log-EI and Thompson sampling (TS). At each step, EI computes the expected improvement upon the best sample so far,

$$\text{EI}(\mathbf{x}) = (\mu(\mathbf{x}) - f(\mathbf{x}^*))\Psi\left(\frac{\mu(\mathbf{x}) - f(\mathbf{x}^*)}{\sqrt{v(\mathbf{x})}}\right) + \sqrt{v(\mathbf{x})}\Phi\left(\frac{\mu(\mathbf{x}) - f(\mathbf{x}^*)}{\sqrt{v(\mathbf{x})}}\right), \qquad (16)$$

where $\Psi(\cdot)$ and $\Phi(\cdot)$ are the CDF and PDF of the standard Gaussian distribution, $\mathbf{x}^*$ is the best sample that gives the largest function value among all the queries up to the current step, $\mu(\mathbf{x})$ and $\sqrt{v(\mathbf{x})}$ are the posterior mean and standard deviation of $f(\mathbf{x})$ given current training data. We used BOTorch to maximize EI through L-BFGS. The most recent work (Ament et al., 2024) introduced log-EI to overcome the numerical challenges in optimizing the original EI acquisition function. Accordingly, we also tested log-EI using the BoTorch implementation. To conduct TS at each step, we used Sobol Sequence (Sobol', 1967) to sample 3K inputs in the domain, and jointly sampled the 3K function values from the GP posterior. We then selected the input with the largest function value as the next query. We denote this method by TS-3K. To explore more input candidates, we also employed the Lancos and conjugate gradient descent to approximate the posterior covariance matrix, and sampled 10K input candidates from Sobol Sequence. We denote this method as TS-10K. We used BOTorch implementation of both methods. We used the Matérn kernel.

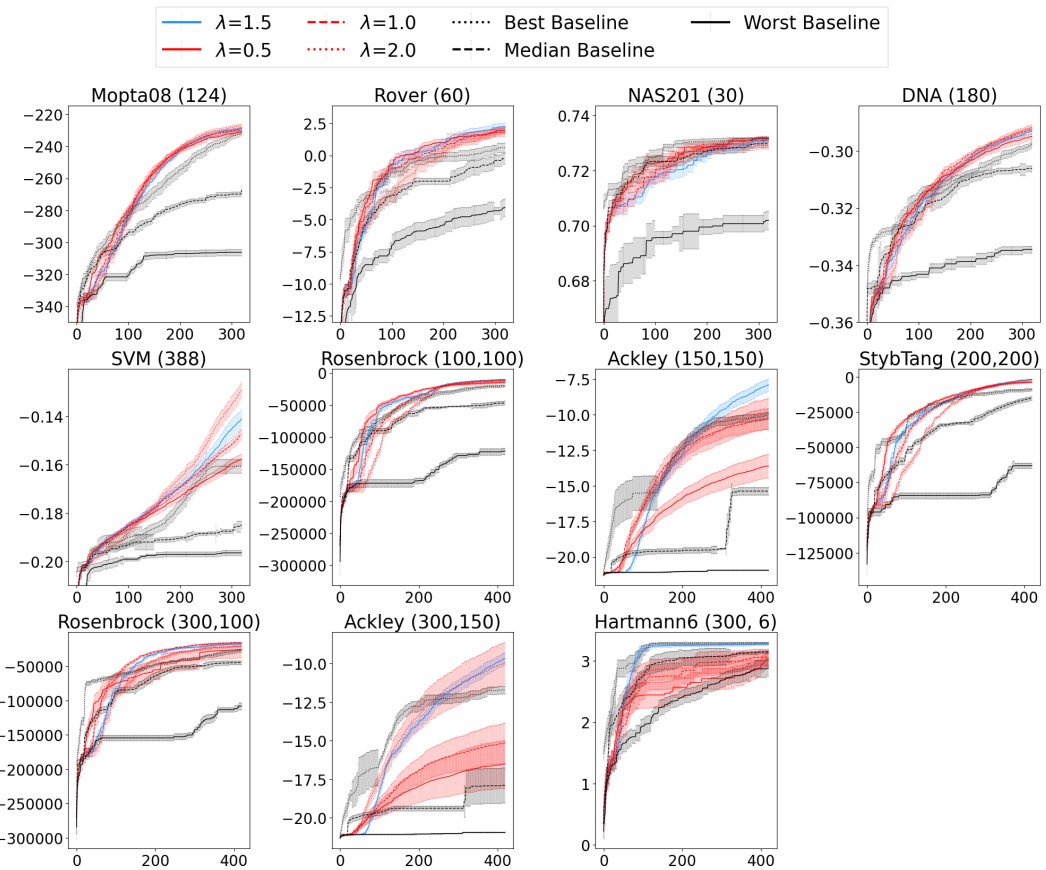

Figure 8: Runtime optimization performance with different exploration level $\lambda$ in UCB. In the main paper and Figure 5 and 6, SBO was compared with other methods using $\lambda = 1.5$.

The ultimate and runtime optimization performance for using EI, TS-3K and TS-10K, are shown in Fig. 9 and 10, respectively. As we can see, in most cases, such as Mopta08 (128), SVM (388), DNA (180) and Rosenbrock (100, 100), both EI and TS perform worse than UCB, and also worse than the best baseline. On Ackley(150, 150) and Ackley(300, 150), the performance of EI and TS are even among the worst. The failure of EI might arise from the gradient vanishing during the optimization, as pointed by the recent work (Ament et al., 2024). The failure of the TS might be due to the high-dimension of the input space. A few thousand candidates might not be sufficient to cover the whole space; as a result, the selected queries are inferior in indicating the best direction to optimize the target function. The use of log-EI performed excellently, achieving results comparable to those obtained with UCB.

# D   Length-scale Initialization Strategies Across Different Gaussian Process Libraries

In this section, we have investigated the initialization strategy of a (non-exhaustive) list of GP training libraries. Most libraries employ a small constant initialization while a few others created the initialization based on the distances between the training data points.

## D.1   Constant Initialization

- Gpytorch[2] initializes the length-scale as *SoftPlus(0.0)=0.693*.

---

[2] https://github.com/cornellius-gp/gpytorch

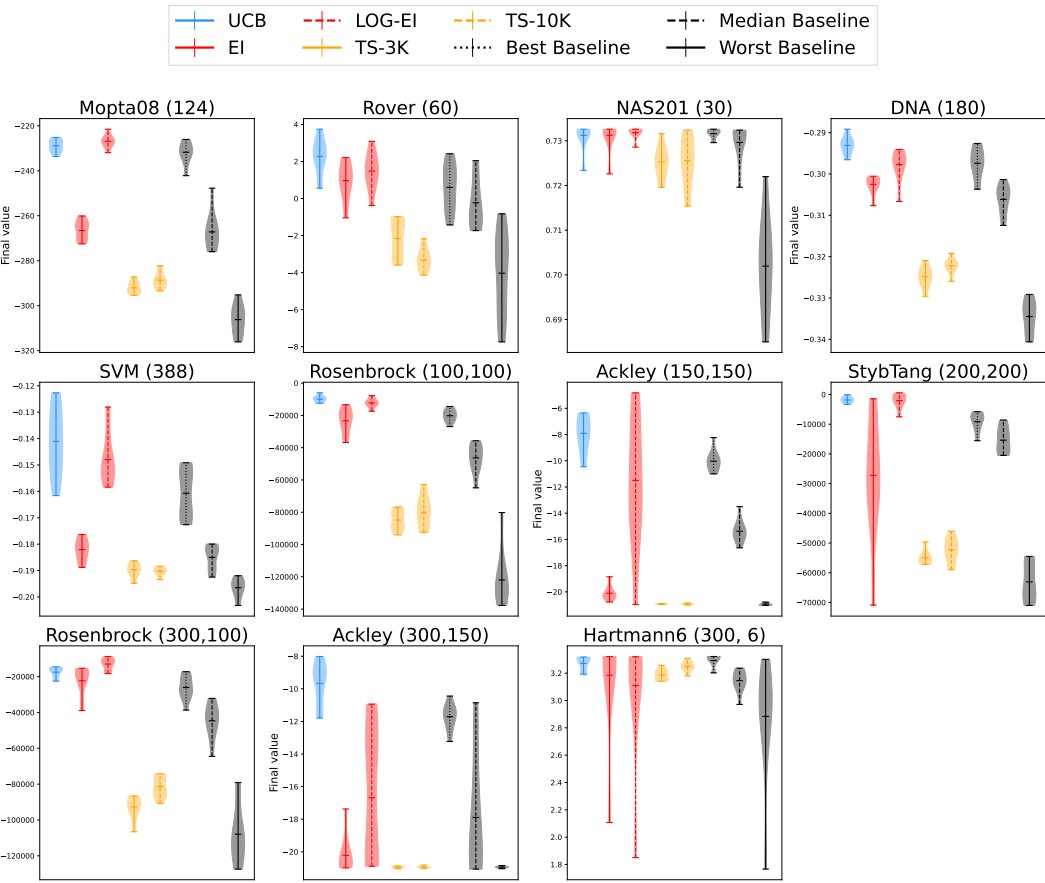

Figure 9: Final performance with different acquisition functions.

- GPy[3] initializes the length-scale as 1.
- Scikit-Learn[4] initializes the length-scale as 1.
- GPML[5] initializes the length-scale as $\exp(0) = 1$.
- GPstuff[6] initializes the length-scale as 1.
- GPJax[7] initializes the length-scale as 1.
- Spearmint[8] initializes the length-scale as 1.
- GPflow[9] initializes the length-scale as 1.

## D.2 Data-Dependent Initialization

- DiceKriging[10] initializes the length-scale randomly based on the range of the input features. The initial length scale for each feature is sampled from a uniform distribution, with a lower bound of $1 \times 10^{-10}$ and an upper bound set to twice the maximum difference of the variable values in the training dataset.

---

[3] https://github.com/SheffieldML/GPy
[4] https://github.com/scikit-learn/scikit-learn
[5] https://github.com/alshedivat/gpml
[6] https://github.com/gpstuff-dev/gpstuff
[7] https://github.com/JaxGaussianProcesses/GPJax
[8] https://github.com/JasperSnoek/spearmint
[9] https://github.com/GPflow/GPflow
[10] https://github.com/cran/DiceKriging

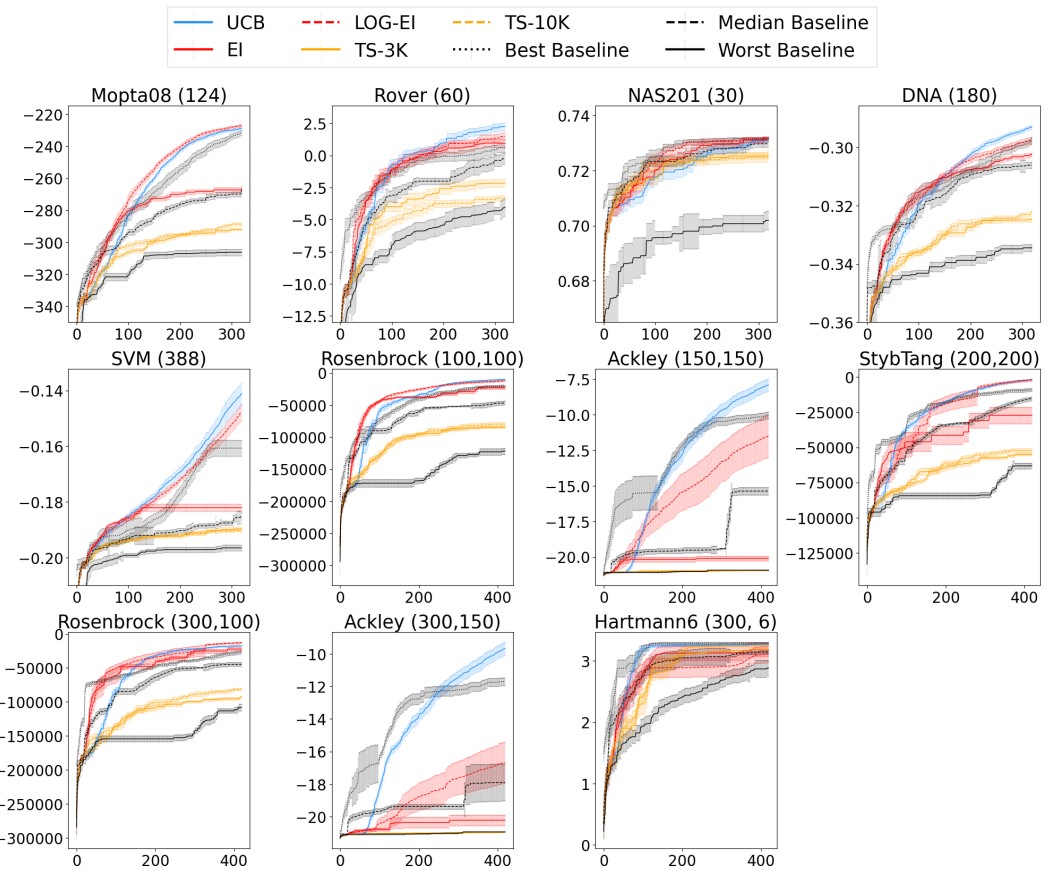

Figure 10: Runtime performance with different acquisition functions.

- hetGP[11] initializes the length-scale to the 50th percentile of the pairwise distances between the training data points.
- GPVecchia[12] initializes the length-scale to be proportional to the mean of the pairwise distances between the training data points.

# E  Limitation

Our current work has identified gradient vanishing as a critical failure mode of SBO in high-dimensional settings. We have proposed an effective, robust initialization method to mitigate this issue. However, we have not yet analyzed whether there is an upper limit to the input dimension for which SBO can function properly, or what that limit might be. To answer these questions, we will continue our investigation in future work, both theoretically and empirically. In addition, we empirically found that the choice of optimizer can also impact GP training performance in high-dimensional settings. For instance, L-BFGS tends to be less stable than ADAM, while RMSProp can outperform ADAM in certain cases. In future work, we will continue exploring initialization strategies to enhance the robustness of GP training across different optimizers.

---

[11]https://github.com/cran/hetGP
[12]https://github.com/katzfuss-group/GPvecchia

