# OpenReview forum: "Standard Gaussian Process is All You Need for High-Dimensional Bayesian Optimization"
_ICLR.cc/2025/Conference — ICLR 2025 Oral_

### Official Review · Reviewer_kZCF · 2024-10-22

**Soundness:** 3
**Presentation:** 3
**Contribution:** 3
**Rating:** 8
**Confidence:** 5

**Summary:**

This paper challenges the widely held notion that Bayesian optimization, using standard kernels, does not scale beyond problems with a, say, single-digit number of independent input variables $d$. The central hypothesis is that standard-kernel GP models _can_ indeed model empirically relevant high-dimensional optimization problems, but are hampered by a problem of hyperparameter tuning. Simply put, if the length-scales $\ell_0$ of isotropic kernels are initialized of order 1, independent of $d$, then the implied covariance between a fixed number of function evaluations drops exponentially with $d$, and so do the gradients of the marginal likelihood, quickly reaching a regime in which hyperparameter optimization becomes numerically impossible.

Instead, the authors propose to initialize kernel length scales as $\ell_0 \sim \sqrt{d}$, and show that doing so effectively counters the exponential decay in correlation between datapoints, and the associated gradients of the marginal likelihood. They argue further that this surprisingly simple change actually suffices to make standard-kernel GP Bayesian Optimization competitive with methods specifically designed to address high-dimensional global optimization problems.

**Strengths:**

This work makes a nice, compact analytical point about about an algorithmic detail, and provides a host of empirical evidence to support the hypothesis. The theoretical analysis appears sound, although the overarching argument can perhaps be questioned (more below). Prior work is extensively reviewed, and the text itself is an enjoyable read.

**Weaknesses:**

Overall I think this is a fine paper. I am not entirely sure I follow the whole argument, though. Here is why:

From my understanding, the argument was never quite that standard kernels are particularly bad as modelling high-dimensional functions, but that general high-dimensional functions are (in the worst case) just exponentially hard to optimize, period. One can "hide" a global optimum among an exponentially growing number of local extrema in, e.g., a sample from a $\ell_0=1$ SE kernel GP (indeed, the analysis provided in this paper indirectly shows that this is true). So, while I completely agree with the authors that the gradients of the marginal likelihood become exponentially suppressed with growing $d$ if the kernel is initialized with $\ell_0=1$, and that this can be alleviated by initializing $\ell_0\sim \sqrt[d}$, a second point also needs to be true for the paper's argument to work: **Practically relevant** objective functions must have some property that constrains their shape such that global optima remain identifiable in feasible sample complexity, *and* that property must be at least approximately captured by the isotropic kernels when initialized in the way proposed in the paper. I am not entirely convinced that the paper provides sufficient evidence in this regard, but I realise that this is a debatable point. Hence I'm trying to phrase a question below.

On a minor note, the authors admit quite clearly that the recent work by Hvarfner et al. achieves pretty much the same effect as their method, but just uses an ever so slightly more involved way to initialize $\ell_0$ by drawing from a LogNormal distribution. Personally, I don't think this is a major issue, since the authors deal with it quite openly. I guess one could argue that this limits the novelty of the work somewhat, though.


### mini issues, easy to fix:
* Section 2, line 091: It's not technically true that BO aims to find new points that are "ideally closer to the optimum". They should just provide helpful information about the location and/or value of the optimium.
* Proposition 4.3 starts with a typo ("Given $\forall$")
* Eq. 6: the parentheses around the argument of $\exp$ are too small. Use `\bigg` or `\left`

**Questions:**

For the moment I have marked the paper as marginally above threshold, but I would be quite willing to raise my score if the point I raised under "weaknesses" is addressed convincingly in the discussion. To re-state it as a question: Are you claiming that practically relevant optimization problems do not have an exponentially growing (in $d$) number of local optima? Or perhaps less explicitly, that practically relevant problems have some other property that means they can be globally optimized with a sample complexity that grows sub-exponentially with $d$? Which part of your work supports this claim? (For example, would you say that the problems listed in Figure 4 are a good surrogate for "practically relevant" problems? How so?

As a counter-example, have you run `SBO-SE (RI)` on a genuine sample of a Matérn $\ell_0=1$ kernel? (Such samples _do_ have an exponentially growing number of local extrema). What happens in such a case?

Thanks in advance for your clarifying reply!


---

Update post discussion:

The authors have addressed my questions. One could always discuss further, and I do see some open questions left here. However, I am convinced this paper provides an empirically highly useful technique, reasonably motivated and supported by theory.

The presence of a related paper does not seem problematic to me, since the related work is amply cited, and the present paper provides a further simplification that users will find highly usable.

Overall, the results presented here will find direct, valuable use with a significant subset of the ICLR community. I thus recommend accepting this paper.

---

> ### Author Response · Authors · 2024-11-18
> **Thanks for the constructive and insightful comments.**
>
> >To re-state it as a question: Are you claiming that practically relevant optimization problems do not have an exponentially growing (in $d$) number of local optima? Or perhaps less explicitly, that practically relevant problems have some other property that means they can be globally optimized with a sample complexity that grows sub-exponentially with $d$? Which part of your work supports this claim? For example, would you say that the problems listed in Figure 4 are a good surrogate for "practically relevant" problems? How so?
>
> R1: Thanks for the insightful questions. We do agree that even there is no training optimization issues, to realistically find the global optimum, the objective function itself must be nice enough and
> additional assumptions must be made, such as the one you mentioned: "practically relevant optimization problems do not have an exponentially growing (in $d$) number of local optima". However, similar to other high-dimensional BO studies, we take a more pragmatic view. That is, we do not require BO to always find the global optimum in practice. In other words, we are open to applying BO to a broad range of problems, even when the global optimum may be challenging to reach. This perspective is common in gradient-based optimization, where, for many applications (e.g., neural network training), a good local optimum suffices for practical use. We believe this expectation broadens BO's practical applicability.
>
> In our experiments, we actually have tested both the nice and tough problems regarding the number of local minima. Among the synthetic benchmarks, **Hartmann6 and Rosenbrock are problems that do *not* have exponentially growing local optima**. Hartmann6 has a fixed dimensionality of 6, while Rosenbrock is unimodal. One can see from Figure 3 and 5, BO successfully converges to the global maximum (3.3237 for Hartmann6 and 0 for Rosenbrock).
>
> In contrast, **Ackley and StybTang exhibit an exponential growth in local optima with the input dimension $d$**. For Ackely, the proliferation of local optima stems from the term $\exp(\frac{1}{d}\sum_{i=1}^d \cos(2\pi x_i))$, and for StybTang, from each summand $(x_i-c_i)^4-16(x_i-c_i)^2 + 5(x_i-c_i)$; see Section B.1 in page 14. Since both objective functions are factorized across dimensions, and each dimension $i$ independently contributes multiple local optima for $x_i$. The combination leads to exponential growth in the total number of local optima with increasing $d$. These functions are more challenging to optimize.   The global maximum is 0 for Ackely and 7833.198 for StybTang. As shown in Figure 3 and 4, after 500 BO iterations, the best function values found are still far away from the global maxima.
>
> >mini issues, easy to fix: Section 2, line 091... Proposition 4.3 ...Eq. 6: the parentheses...
>
> R2: We greatly appreciate you pointing out these issues. We will address and correct them in our manuscript.

---

> > ### Comment · Reviewer_kZCF · 2024-11-18
> > **Thanks for the clarifications**
> >
> > Dear authors,
> >
> > thanks for the clarifications! I have to admit I didn't recall that Ackley and StybTang indeed have an exponentially growing number of local optima.
> >
> > Do we agree, then, that your approach amounts to a deliberate model mismatch that primes the Bayesian optimizer towards more exploration, even if this can lead to the model missing the global optimum for a potentially longer time (as in your experiments on the aforementioned two test problems)? In other words: Increasing the length-scale with as a function of the domain dimensionality does not improve the *model* per se -- in fact it may lead to model mismatch, for example for objective functions whose optimization complexity grows exponentially with dimensionality. But this tactic changes the behavior of the optimizer in a way that seems beneficial on your benchmarks.
> >
> > If you are you planning to make this aspect more explicit in the paper, I would be supportive. This seems like an important point to clarify the message of the paper.

---

> > > ### Author Response · Authors · 2024-11-19
> > >
> > > Dear Reviewer:
> > >
> > > Thanks for your insightful questions. To clarify, what we proposed --- "a function of the domain dimensionality $d$" --- is merely an  **initialization** of the length-scales, rather than the final values of the length-scales used to conduct each BO step. After this initialization, the GP training process will further *adjust* the length-scales to well fit the data. Since GP training objective is typically non-convex, there is indeed a risk that, with our proposed initialization, the trained model may not perfectly match the true underlying function. But an effective data-driven training process is still likely to identify a reasonably good approximation. This mirrors the famous quote: "All models are wrong, but some are useful".
> > >
> > > Our main argument is that, with our proposed initialization, the training process is much less likely to start with vanishing gradients, which **largely improves the chances of successfully training the model and obtaining a useful, albeit potentially imperfect approximation**.
> > >
> > >
> > > In contrast, consider using a constant initialization (invariance to $d$), as is commonly done. First, when the function is really complex, say, with exponentially growing local minima, there is no guarantee that such an initialization will lead to a better approximation of the target function than starting with our proposed initialization. More importantly, this initialization is much more likely to result in **vanishing gradients at the start of training, preventing any effective updates throughout the training process**. This failure can produce a very poor approximation of the target function, which in turn hinders effective search and querying in BO steps, ultimately leading to poor optimization results. This has been observed in our experiments with Ackley and StybTang (see Figure 3).

---

> > > > ### Comment · Reviewer_kZCF · 2024-11-21
> > > > **Thanks. Questions resolved, score increased.**
> > > >
> > > > Thanks, this helped me understand better.
> > > >
> > > > Thanks for the professional discussion! I've increased my score.

---

> > > > > ### Author Response · Authors · 2024-11-22
> > > > >
> > > > > Thank you! We appreciate your response.

---

### Official Review · Reviewer_VLW7 · 2024-10-22

**Soundness:** 3
**Presentation:** 2
**Contribution:** 3
**Rating:** 8
**Confidence:** 4

**Summary:**

The common assumption is that Gaussian Process-based Bayesian Optimization inherently scales poorly with the dimensionality (D) of the search space, such that the search is virtually uninformed for D surpassing 15-20. This paper investigates the validity of this assumption and the reason for the typically observed poor performance on search spaces of high dimensionality. They show that, using the popular SE-kernel, the gradient of the log likelihood with respect to the lengthscales vanishes (with high probability) in the case of high dimensionality and typical initial lengthscales, preventing effective gradient-based training of the lengthscales. While Matérn kernels also experience vanishing gradients, the problem occurs at much larger D compared to SE, which aligns with their experimental performance results. Furthermore, the authors show that initialising the lengthscales to \sqrt{D} is sufficient to overcome the vanishing gradient issue and achieving performance that matches or exceeds that of more advanced Bayesian Optimization methods.

**Strengths:**

The overall story of the paper is easy to follow, the investigated topic is important, and the proposed approach is sound. The theoretical analysis (Section 4) of the vanishing gradients is interesting and, as far as I am aware, presents a novel perspective on the curse of dimensionality in this context.

**Weaknesses:**

The central thesis of this paper is that a standard "no bells and whistles" implementation of Gaussian Process (GP)-based Bayesian Optimization (BO) can perform competitively with state-of-the-art methods for high-dimensional BO, requiring only a straightforward adjustment to common practices. However, this thesis closely mirrors that of the existing work (1) "Vanilla Bayesian Optimization Performs Great in High Dimensions" (ICML 2024), which presents a similar argument.

In the paper under review, the authors propose scaling the lengthscales proportionally to \sqrt{D}, and they achieve this by initialising the lengthscales with \sqrt{D}. The existing work (1) similarly advocates for scaling the lengthscales in proportion to \sqrt{D}, as discussed in Section 5.1 of their paper. They suggest implementing this adjustment by setting the hyperparameters in the prior on the lengthscale appropriately, noting that using a maximum a posteriori (MAP) estimate for a hyperprior is a straightforward and widely-used approach.

While the reviewed paper does cite the existing work, its claim that the proposed method is "even simpler" (see Section 5 "Alternative Method") is not entirely convincing. Both papers recommend favouring lengthscales near \sqrt{D}; the difference lies in how they implement this: the reviewed paper achieves it solely through initialisation, whereas the earlier work suggests adjusting the prior on the lengthscales. As it stands, the reviewed paper presents the main thesis as though it were novel (at least in the abstract, introduction and conclusion), despite its overlap with the prior work.

Nevertheless, there are valuable contributions in the reviewed paper, such as the analysis of vanishing gradients, which offers an insightful perspective on the consequences of using inappropriately small (but typically used) lengthscales. To improve the work, the authors could clearly position it as an expansion upon the thesis established in the earlier paper, emphasising the novel aspects of their analysis and findings.

**Questions:**

I do not have any questions. Please see the suggestions on how the work could be improved under "Weaknesses".

---

### Official Review · Reviewer_RmNw · 2024-10-30

**Soundness:** 2
**Presentation:** 2
**Contribution:** 2
**Rating:** 6
**Confidence:** 3

**Summary:**

This paper thoroughly investigates the performance of standard GP in high-dimensional BO. Specifically, the paper focuses on the use of SE and Matern kernels in standard GP, along with analysis on their gradient values during GP training. The authors point out the failure of using SE kernel is mainly due to the vanishing gradient problem and propose a new initialization strategy for kernel length-scale that mitigates the problem and improves performance. Comprehensive experiments are carried out in comparing performance of the proposed standard BO (SBO) against other state-of-the-art BO algorithms on various optimization tasks.

**Strengths:**

1. The authors claim the vanishing gradient in the SE kernel training is the main cause of BO failure in high-dimensional cases and provide detailed theoretical analysis on the gradient values of SE kernel. The theoretical works are easy to follow and provide insights on how likely the vanishing gradient problem can occur when initial values of length-scale are fixed in high-dimensional setting. The proposed initialization strategy is also supported by a probabilistic bound.

2. The authors perform adequate experiments to validate their statement. Both high-dimensional synthetic functions and real-world applications are considered, and the chosen BO algorithms are representative and classic in the field of high-dimensional BO. In addition to theoretical analysis, the authors also provide several experiments results to confirm both the existence of vanishing gradient problem and its relationship with initial length-scale and dimensionalities.

**Weaknesses:**

1. This paper mainly focuses on evaluating existing method. Vanishing gradient problem has been widely discussed in gradient-based training. I think the novelty of this submission only comes from the proposed initialization method, but the authors spend only limited paragraph highlighting the performance of this proposed method. In the real-world problems, the proposed SBO-SE(RI) method even worsens the performance of the original SBO-SE in most cases (namely every case except for the SVM(388) according to Figure 4).

2.  I'm not convinced by the experiments design in Section 4, where the authors try to further validate their theoretical results by varying $l_0$ and $d$. (1) The evaluation metric used to confirm the occurrence of vanishing gradient lacks generality. To my understanding neither the $L_2$ difference nor the average gradient norm at the first training step reveals the presence of vanishing gradient throughout the training process. The number of training steps is not specified by the authors but it is safe to say there are more than 1 step of kernel parameter training. Therefore, the numerical difference in $L_2$ norm of length-scale before and after training only proves $l$ was updated in some steps but vanishing gradient could still have occurred in other steps. The latter gradient norm metric is only considered for the first step of training without a generalization to the whole training process. Vanishing gradient could still occur in the following steps. Step-wise gradient/length-scale evaluation would be more convincing. (2) The values of length-scale initialization ($l_0$) is not robustly tested. In line 199-200, the authors state fixed $l_0$ will lead to vanishing gradient and in the later numerical verification (line 239-240) several fixed values of $l_0$ are tested. However, given the minimal chosen value of $d$ is $50$, $\sqrt{d} > 7$ which is much larger than other choices always. For better generality, I think some larger fixed value of $l_0$ (e.g. $5, 15, 25$) should also be considered as a stronger support for the adaptive $l_0 = c\sqrt{d}$ setting.

3. The readability of this submission needs to improve. Starting from line 233, I keep being instructed to jump between sections to get details and results of the experiments. For example, in line 236 I need to go to section 6.1 to get definitions of the benchmark notations. Similarly in line 264 I'm referred to Figure 3 and 4 which are at the very end of the paper. It would be much easier to read if the authors could gather all the numerical experiments into Section 6 instead of alternating between theoretical presentation and experiments evaluation (e.g. move line 233-324 to the start of section 6).

**Questions:**

1.  In equation (1) and (2), both ARD SE kernel and ARD Matern kernel contain the amplitude $a$. Since in line 405 a Gamma prior is placed on this parameter, I assume the authors also include $a$ as the trainable parameter in GP training. How did the authors initialize $a$ in GP training? In equation (6), is the amplitude $a$ considered as a constant during theoretical analysis? In line 209-212, why is $a$ not presented in the gradient expression of the Matern kernel?

2.  In Figure 2, the relative length-scale differences for both GP-Matern($l_0=0.693$) and GP-SE($l_0=\sqrt{d}$) are flat lines. Since $l_0$ is initialized at the same value independent of the number of iteration, does this mean the value of length-scale is all the same after training even for different GP (since the GP at each iteration is modeled on different data sets)?

3.  Consider cases: Ackley(150, 150), Mopta08(124), Rover(60), NAS201(30), DNA(180) in Figure 3 and 4 (5 out of the 11 benchmarks), it seems adding the robust initialization to SE kernel worsens its performance. How would the author justify this observation supports their statement in line 345-346 ("the performance across all the benchmarks is dramatically improved")?

---

> ### Author Response · Authors · 2024-11-18
> **Response**
>
> We thank the reviewer for their time and detailed feedback. We are concerned, however, that certain key aspects of our work --- including its primary contributions, motivation, results, and the interpretation of our gradient vanishing experiment --- may **not** have been fully understood. We provide the following clarifications to address these points in detail.
>
> > This paper mainly focuses on evaluating existing method. Vanishing gradient problem has been widely discussed in gradient-based training. I think the novelty of this submission *only comes from the proposed initialization method* ...
>
> R1: We respectfully disagree.
>
> First, our paper does **not** "mainly focus on evaluating existing methods." As clearly summarized in the Introduction (Lines 58-81), this work includes three main contributions: **Empirical Results**, **Theory**, and **Simple Robust Initialization**. While we conducted extensive empirical evaluations, these form just one part of our contributions. The theoretical analysis and its numerical verification are central to our study, occupying significant space: the entirety of Section 4 (pages 4-6), two tables (Tables 1-2), two large figures (Figures 1 and 2), and all of Section A in the Appendix. **We are uncertain as to why this substantial portion of the work was overlooked by the reviewer.**
>
> Second, we were surprised to see our theoretical analysis is dismissed as **lacking novelty or contribution** on the grounds that **"the vanishing gradient problem has been widely discussed in gradient-based training"**. This critique seems overly reductive and sloppy. By that logic, the vast body of contemporary neural network research would also lack novelty simply because neural networks have been widely discussed since the 1980s.
>
> To clarify, our contribution is **not** the concept of "gradient vanishing" itself. Rather, we identified why standard BO with a SE kernel often fails in high-dimensional problems --- specifically, because training fails due to gradient vanishing right from the initial training step. This discovery is non-trivial, as BO is a complex, iterative process involving interleaved GP training and acquisition function optimization. We not only identify that the vanishing gradient arises from an inappropriate yet commonly used length-scale initialization, but also provide a probabilistic bound that characterizes the likelihood of gradient vanishing, which grows rapidly with the dimensionality, $d$. **To our knowledge, the identification of this failure mode in BO and the accompanying quantitative analysis are the first in this area, and therefore represent novel contributions of our work**.

---

> ### Author Response · Authors · 2024-11-18
> **continue**
>
> >I'm not convinced by the experiments design in Section 4 ... To my understanding neither the $L_2$ difference nor the average gradient norm at the first training step reveals the presence of vanishing gradient throughout the training process ... Vanishing gradient could still occur in the following steps.... The values of length-scale initialization ($\ell_0$) is not robustly tested...
>
> R2: There appears to be a misunderstanding about the motivation and design of our experiments. To clarify, the theoretical analysis in Section 4 is aimed at demonstrating that the gradient vanishing can occur **at the very first step of training** due to an inappropriate length-scale **initialization**. When gradient vanishing occurs at this initial step, it persists throughout training, as each subsequent step continues to compute gradients at the same initial values, preventing any updates to the length scales and ultimately causing training failure. Thus, we specifically use (1) the gradient norm at the first step to examine whether gradient vanishing occurs initially, and (2) the relative $L_2$ difference between the length-scales before and after training to confirm that, once gradient vanishing happens at the first step, it persists across all training iterations, resulting in no change after training. **Our paper jointly interprets the results of (1) and (2) to verify our theory, rather than treating these metrics independently**.
>
> In other words, our experiments are **not** designed to assess whether gradient vanishing might occur later in the training process. In such cases, **the length-scales would already be effectively updated (before any gradient vanishing), meaning training would not necessarily fail**.
>
> Second, we employed both commonly used length-scale initialization values and our proposed values for verification. The results sufficiently demonstrate that smaller values are more prone to gradient vanishing and training failure, while larger initializations handle higher dimensions robustly (our results are averaged over 20 runs); see Lines 245-260.
>
> Moreover, with $\ell_0 = 1.0$, there has already been no gradient vanishing observed for the Matern kernel across all the test dimensions. This suggests that even larger values, such as $\ell_0 = 2$ or $3$, would also prevent vanishing gradients across all dimensions, making it unnecessary to duplicate results with larger initializations. While further testing with additional values is feasible, we believe our current test range is sufficient to verify our theoretical analysis.
>
> >Questions regarding $a$ in training and analysis
>
> R3: Thank you for these questions. In our experiments, the initialization value of $a$ was set to $\text{SoftPlus}(0) = 0.693$, which is the default choice in the BoTorch library. We will clarify this in the paper. There is a typo in the gradient equation for the Matern kernel, where $a$ was omitted; $a$ should be a factor. We will correct this. Lastly, yes, the amplitude $a$ is treated as a constant during the theoretical analysis.
>
> >In Figure 2, the relative length-scale differences for both GP-Matern($\ell_0=0.693$) and GP-SE($\ell_0=\sqrt{d}$) are flat lines. Since $\ell_0$ is initialized at the same value independent of the number of iteration, does this mean the value of length-scale is all the same after training even for different GP (since the GP at each iteration is modeled on different data sets)?
>
> R4: No. The figure just shows that the overall change of the length-scale values,
>  $L_2=\frac{|| \ell_1 - \ell_0||}{||\ell_0||}$, is close across the BO iterations, but it does not mean the actual learned values are the same. For example, starting from $\mathbf{\ell}_0 = [1,1,1]$, consider two sets of the learned length-scales  $\mathbf{\ell}_1 = [0.7, 0.6, 0.3]$ and $\mathbf{\ell}_2 = [0.5, 0.505, 0.505]$. They are obviously different, but the relative $L_2$ difference from the initialization is almost the same:  $||\mathbf{\ell}_1 - \mathbf{\ell}_0|| = 0.86023$ and $||\mathbf{\ell}_2 - \mathbf{\ell}_0|| = 0.86026$.

---

> > ### Author Response · Authors · 2024-11-18
> > **continue**
> >
> > >Consider cases: Ackley(150, 150), Mopta08(124), Rover(60), NAS201(30), DNA(180) in Figure 3 and 4 (5 out of the 11 benchmarks), it seems adding the robust initialization to SE kernel worsens its performance. How would the author justify this observation supports their statement in line 345-346 ("the performance across all the benchmarks is dramatically improved")?
> >
> > R5: This is a typo --- the word "failed" was inadvertently omitted. The sentence should read: *As shown in Fig. 3 and Fig. 4, the performance across all the **failed** benchmarks is dramatically improved*. We will correct this in the paper.
> >
> > We would also like to emphasize that our method is designed specifically to prevent the failure mode often encountered by standard BO (SBO) with the SE kernel. **Our goal is not to outperform SBO in cases where it functions well with standard initializations but rather to provide robust performance where SBO commonly fails**. The results confirm that our method achieves this goal: when SBO with commonly used initializations fails, our approach enables SBO to reach, or approach, state-of-the-art performance. In cases where SBO performs well with standard initializations, our method may occasionally perform slightly less optimally, but these differences are minor, and our results remain among the best.
> >
> > We believe it is neither realistic nor necessary to expect a method to be the absolute best in every context and benchmark.

---

> ### Comment · Reviewer_RmNw · 2024-11-22
> **Response to authors**
>
> Thank you for the clarifications. I was a bit disappointed previously because when the authors state they are paying attention to the issue of gradient vanishing, I was expecting a more general approach to resolve this problem for the entire training process. I was also misled by the typo made in the authors' comment for their experiment results. Now after the authors' further clarification on the conclusion of the experiments and the scope of the problem they are studying, I acknowledge the novelty and contribution of this work and increase my score.
>
> I would like to make two more comments:
> 1. The phrase $\textbf{Runtime performance}$ (line 448) is misleading. Readers will expect to see actual running time reported. The authors might consider using a different word.
> 2. The authors used a very general title for their submission which gives readers impression that the challenge in high-dimensional BO has been solved by standard GP (also one of the reasons why I was expecting a lot previously $\textbf{:(}$ )

---

> > ### Author Response · Authors · 2024-11-24
> >
> > Thank you! We appreciate your response. Also thanks for  highlighting places that could benefit from clarification. We will address the typos and rephrase “Runtime performance” to improve clarity. Additionally, we will refine our draft to better highlight the objectives and scope of our work.

---

### Official Review · Reviewer_pG21 · 2024-11-04

**Soundness:** 3
**Presentation:** 4
**Contribution:** 4
**Rating:** 8
**Confidence:** 4

**Summary:**

The authors investigate vanishing gradient problems associated with the squared exponential and Matern kernel functions, finding them both to exhibit gradient vanishing at common default initializations. They propose a dimension-dependent initialization and advocate for the Matern kernel, and demonstrate on numerical experiments that this can actually be more important than all the bells and whistles we have been putting on GPs for the last decade.

**Strengths:**

The aim of this work, to challenge a basic, un-sexy component of the BO framework as the lengthscale initialization/covariance kernel and to attempt to pin down the influence on model performance is great.

At first, I found it difficult to fully take in the implications. But if my quick refreshing/skim of them is to be trusted, the articles proposing TuRBO, Tree UCB, RDUCB, HESBO, ALEBO, SaasBO do not have any "vanilla BO" baseline: they compare only to other sophisticated methods. This article may have revealed that the emperor has no clothes: this thread of literature may indeed have been missing the basic Matern kernel as a suitable approach to high dimensional BO. If correct, this finding will have a significant impact on the future of BO research in high dimension. It should probably also prompt some soul-searching among us who have been doing work in this area.

**Weaknesses:**

---

Insufficient replicates in the numerical experiments.

Can a study with 10 replicates really be said to be a "comprehensive evaluation"? Particularly when there is such a large variance in the results: maybe half of the method-problem combinations seem to span the entire plot. Given how influential the conclusions of this article would be if correct, it is really disappointing that they are based on estimates of performance with such high standard error. Running (significantly) more experiments would solidify the important findings of this article. Really, there should probably be closer to 50 replicates or more.


---
Criticism of the squared exponential and advocacy of the Matern kernel is not new.

Stein, M. L. in Interpolation of Spatial Data makes this argument in 1999, and that is referenced in the covariance chapter of Rasmussen and Williams.

It is probably worth citing at least these documents, and trying to locate other discussion of this topic.

---

While there are indeed (surprisingly) many software platforms which use a constant lengthscale initialization (including GLflow, GPJax, Spearmint, GPStuff, GPML, scikit-learn and GPy, based on my quick skim of their source code), this is not universally the case.

There are also packages that do indeed use the idea of involving the expected distances between points when choosing the lengthscales.
These packages actually do so empirically instead of assuming a uniform input distribution, which is surely violated in, say, Bayesian Optimization settings. The ones I found based on a quick search were:
i) DiceKriging uses multiple random initializations between 1e-10 and twice the range of the distances.
ii) hetGP initializes the lengthscale based on the quantiles of the observed distances.
iii) GPVecchia seems to use 1/4 times the mean distance between training points.

It is probably worth mentioning these and situating them relative to your proposal.

**Questions:**

Why not use a Matern kernel with the proposed initialization in your numerical experiments?

Under the assumption of uniform sampling, does any of the procedures of DiceKriging, hetGP or GPVecchia satisfy your robust initialization?

---

> ### Author Response · Authors · 2024-11-18
> **Thank you for insightful and constructive comments**
>
> We thank you for your support, and the many insightful and constructive comments.
>
> >Insufficient replicates in the numerical experiments....Really, there should probably be closer to 50 replicates or more.
>
> R1: Thank you for the great suggestion; we agree. As recommended, we will add more replicates --- around 40 additional runs or more --- to further reduce variance and strengthen our findings.
>
> >cite documents about "Criticism of the squared exponential and advocacy of the Matern kernel" and "trying to locate other discussion of this topic."
>
> R2: Thank you for the helpful suggestions! We will add the provided reference along with additional relevant literature on this topic. We will also expand our discussion of this point.
>
> >Why not use a Matern kernel with the proposed initialization in your numerical experiments?
>
> R3: Great question. We empirically found when Matern kernel with commonly used initialization does not exhibit  gradient vanishing and training failure, its performance is close to that achieved with our robust initialization. Since the Matern kernel with commonly-used initialization performs well across all benchmarks in the paper (without gradient vanishing), adding additional results with our new initialization would make the figures overly cluttered --- particularly in Figures 3, 4, and 5, where numerous baselines have already been included --- and would reduce readability.
>
> Here, we have included a comparison of the Matern kernel using both initializations on the Rover, DNA, SVM, and Hartmann6 benchmarks. The results are shown in [final](https://drive.google.com/file/d/1uflD7GGlV5gmFBA89q6O5ROLZQsc_GWz/view) and [running](https://drive.google.com/file/d/1WX1hdtdBnw7XpoUHxAqmBn4w6eoxUd1I/view)
> , demonstrating that the performance is indeed very similar between the two initializations. We will clarify this point in the paper.
>
> We have also conducted an additional test on a new benchmark, *Humanoid*, with a dimensionality of $d = 6,392$. Please refer to our response to @Reviewer M8nK for the results. In such high dimensions, we observed that commonly used initialization leads to gradient vanishing and training failure even for the Matern kernel. However, our robust initialization approach dramatically enhances performance, achieving state-of-the-art results.
>
> >Under the assumption of uniform sampling, does any of the procedures of DiceKriging, hetGP or GPVecchia satisfy your robust initialization?
>
> R4: Thanks for providing these excellent GP learning packages. We appreciate the opportunity to compare our initialization approach with those used in *DiceKriging*, *hetGP*, and *GPVecchia*.
>
> For *DiceKriging*, the initializations are independently sampled across input dimensions, with values ranging between $1 \times 10^{-10}$ and twice the maximum possible value in the domain (in our case, this is $[1 \times 10^{-10}, 2]$). Therefore, these initialization values are independent of the input dimension $d$ and do not align with our approach.
>
> In *hetGP* and *GPVecchia*, the initialization strategies rely on the training data, making them data-dependent. However, in expectation, their initializations can match our approach. For *hetGP*, the initialization can be set to the 50th percentile of the observed pairwise distances. This provides a stochastic estimate of the median pairwise distance, which is on a similar scale to the mean pairwise distance. In our analysis, we assume a uniform data distribution within $[0, 1]^d$, where the mean pairwise distance between data points is approximately $\sqrt{d}/6$. Thus, the *hetGP* initialization is, on average, consistent with our robust initialization.
>
> Similarly, in *GPVecchia*, the initialization is a proportion of the mean distance between training points, which provides a stochastic estimate of the mean distance scaled by a constant. This, too, aligns with our robust initialization strategy in expectation.
>
> We will add a discussion of these methods and the similarities and differences in our paper.

---

> > ### Comment · Reviewer_pG21 · 2024-11-26
> >
> > Cool, I think even if there's not a big empirical difference for the Matern if scaled or not it's still important to show so that's great that you have that.
> >
> > Thanks for looking into those other packages' initializations and I think this will really help the article be better absorbed by the field.

---

> > > ### Author Response · Authors · 2024-11-28
> > >
> > > Thank you for your response. We completely agree and will incorporate these results, along with discussions on initialization strategies from various packages, into our manuscript.

---

### Official Review · Reviewer_M8nK · 2024-11-04

**Soundness:** 4
**Presentation:** 3
**Contribution:** 3
**Rating:** 8
**Confidence:** 3

**Summary:**

This paper considers the problem of high-dimensional Bayesian Optimization with Gaussian Processes, in particular the selection of a kernel. It argues that the poor performance of the standard SE kernel in high dimensions stems from the vanishing gradient problem, and provides theoretical and experimental evidence to this effect. It then proposes a method to initialize the parameters of the SE kernel (setting the length-scale parameters to $O(\sqrt{d})$) to tackle this problem. Experimental results on synthetic and real datasets show that this initialization brings up the performance of the SE kernel to be comparable to the state-of-the-art.

**Strengths:**

- The paper is well-written, clear, and straightforward to understand.
- Hyperparameter optimization is a problem with wide-ranging applications.
- As far as I know, the consideration of the gradient vanishing problem for Bayesian Optimization with Gaussian Processes is novel. It is quite interesting, and indicates that it might be a pertinent direction of analysis for other applications due to the popularity of gradient descent.
- The paper provides both experimental and learning theoretic evidence of the gradient vanishing problem as the problem dimension increases.
- The proposed fix brings up the performance of standard BO with GPs and classic SE kernel to be comparable to the state-of-the art, including on NAS201.

**Weaknesses:**

- The dimensionality of the problems in the experiments are fairly high, but some problems in neural architecture search have even higher dimensionality.
- The proposed method does not outperform the baselines. However, I don't see this as a major weakness since I feel the main message of the paper is not the proposed method, but the importance of considering the gradient vanishing problem.

**Questions:**

Have you done experiments on problems with even higher dimensions, e.g. in the thousands?

---

> ### Author Response · Authors · 2024-11-18
> **Thanks for your support and great suggestions.**
>
> >Have you done experiments on problems with even higher dimensions, e.g. in the thousands?
>
> R1: While our benchmarks are drawn from widely used and recognized sources in high-dimensional optimization literature, we agree that testing in even higher dimensions, such as in the thousands, is valuable. To address this, we conducted an additional test on the Humanoid benchmark, a reinforcement learning task with a dimensionality of 6,392, as used in recent work by Hvarfner et al., (2024). We ran standard BO with both Matern and RBF kernels using MLE estimation and ADAM optimization over 1,000 BO iterations.
>
> The results are available [here](https://drive.google.com/file/d/1aC1GendhntPpB_k0eZQ54sApidSRz-fx/view). In this high-dimensional setting, we observed that commonly used initializations lead to gradient vanishing and training failure for both kernels. With our robust initialization method, however, performance for both kernels improved dramatically, achieving results close to VBO (Hvarfner et al., 2024), which employs a log-normal prior for training.
>
> We will supplement our paper with these results on the Humanoid benchmark.

---

> > ### Comment · Reviewer_M8nK · 2024-11-27
> >
> > After reading the other reviews and rebuttal, I have decided to keep my score.

---

> > > ### Author Response · Authors · 2024-11-28
> > >
> > > Thank you! We appreciate your response.

---

### Comment · Reviewer_pG21 · 2024-11-14

Thanks Authors for this interesting work and Reviewers for sharing your insights.

I have learned much from reading the reviews of my fellow Reviewers, and wanted to create a thread here dedicated to discussing the impact of Hvarfner et al 2024 on the novelty of the article presently under review, as mentioned by reviewers VLW7 and kZCF. In my updated view, adjudicating this is of significant concern for determining whether this article is to be recommended for publication (though I understand there are additional concerns, RmNw's among them).


I would like to submit two points for discussion on this matter.


---

1.
Looking at the dates, the work under review and Hvarfner et al 2024 were initially posted on arXiv within two days of one another. Consequently, there is an argument to be made that this is concurrent work.


---

2.
Contra VLW7, I think my view is that it is indeed nontrivially simpler to suggest an initialization rather than a prior distribution to accomodate the $\sqrt{d}$ scaling. It's true that the prior recommended by Hvarfner et al is getting us to the same place, and doesn't involve any extra hyperparameters, but the requirement to use a prior comes with more metaphysical commitments relative to just setting an initialization. And what if I already had some different prior in mind that I wanted to use?

But more importantly, I think it is even simpler still to just suggest using a Matern kernel rather than a SE. I think that the conclusion that "We would have been better off just using the Matern kernel than doing any of the fancy things we've developed over the last decade and a half" is an extremely powerful one.

---
But I'm very interested in hearing input from my fellow Reviewers as well as the Authors on this matter.

---

> ### Comment · Reviewer_VLW7 · 2024-11-16
>
> Like pG21 I wish to thank the authors for the interesting work and the reviewers for their insights.
>
> @pG21: Thank you for your comments. Let me expand on some points and clarify further.
>
> 1. That is indeed an excellent point --- thank you for highlighting it, I was not aware of this. With that in mind, there is certainly a case to be made for considering it concurrent work.
>
> 2.
> I think you're absolutely right that the notion of "simpler" is important here, however, it can be interpreted in different ways depending on context.
>
> I agree that it can sometimes be simpler for a practitioner to implement it this way.
>
> However, the approach of "encouraging" a solution via its initialization primarily applies when employing local optimization methods for hyperparameters, such as gradient descent or Adam. It does not generalize well to global optimization algorithms. Defining an appropriate lengthscale prior is much more general, and in combination with other things, can often make things simpler rather than more difficult. This approach explicitly embeds the desired properties into the optimization objective for the lengthscale, making it adaptable to a wide range of optimization methods (local or global).
>
> When relying solely on initialization to encourage a solution, you're implicitly assuming that the algorithm's movement away from the initial point will be limited. This assumption can become problematic. If the optimization algorithm is highly effective and explores globally optimal solutions, it might counteract the initialization's influence, potentially returning to less desirable lengthscales (which can be better according to the objective/loss function). This is particularly true for global optimization algorithms (which are popular and useful in these settings), where the influence of initialisation tend to be much weaker. To mitigate this, you might need to constrain the algorithm artificially (e.g., limiting exploration or stopping optimization early), which can add complexity rather than reduce it.
>
> By contrast, modifying the objective function for the lengthscale optimization provides a clear, unambiguous goal, and is easily done via the hyperparameter prior. This approach aligns the optimization process with the desired outcome, which in turn can lead to a more robust system and a simpler experience for the practitioner.
>
> Moreover, in BO it is common and often advantageous to rely on Bayesian inference, to consider a (data agreement-weighted) range of hypotheses for the hyperparameters, instead of relying on point estimates [1]. In this context, a well-defined prior is far more practical than relying on implicit biases introduced through initialization. The prior integrates seamlessly with inference procedures, avoiding the need for algorithmic workarounds.
>
> "What if I already had a different prior in mind that I wanted to use?"
> This is a great question! The prior is simply a density function. Importantly, it doesn't need to be normalized for optimization or Bayesian inference, as the normalization constant often cancels out during these processes. This flexibility makes it easy to combine or adjust priors to reflect different assumptions or preferences.
>
> My point here really is that encouraging a solution via its initialization _is_ changing the prior, it just means doing so in an implicit way, where you also need to consider how the optimization/inference algorithm works. Sometimes this is a practical, pragmatic approach which makes things simpler to implement, but it has its limitations, and can make things harder instead.
>
> In my opinion, now considering this concurrent work, I think the paper can be made ready for publication by merely smaller edits with regards to the other paper. In particular, under "Alternative method" (line 353-360): (1) making it clear that it is concurrent work while highlighting differences in focus, and (2) remove/lessen the argument about it being "simpler", which argued above, is needlessly ambiguous, and very context specific.
>
> [1] Snoek, Jasper, Hugo Larochelle, and Ryan P. Adams. "Practical bayesian optimization of machine learning algorithms." Advances in neural information processing systems 25 (2012).

---

> ### Author Response · Authors · 2024-11-18
> **Thank you for your detailed and constructive comments**
>
> Thank you for your detailed and constructive comments. We would especially like to thank @Review pG21 for clarifying that our work is concurrent with (Hvarfner et al., 2024) and highlighting the differences between our approach and theirs. We will incorporate these clarifications into our paper.
>
> >The difference from (Hvarfner et al 2024).
>
> R1: We agree that both (Hvarfner et al., 2024) and our work demonstrate that standard (or vanilla) Bayesian Optimization (BO) can perform effectively in high-dimensional optimization. Below, we provide a detailed summary of our unique contributions:
>
> 1. We identify that the failure of standard BO, particularly with the SE kernel, stems from training failure caused by gradient vanishing. The gradient vanishing arises from commonly used but inappropriate initialization.
>
> 2. We provide a mathematical analysis of the gradient vanishing issue, presenting several bounds, including a probability bound, and characterize its relationship with the input dimension $d$. Numerical verification further supports our analysis.
>
> 3. We find that the Matern kernel is a more robust choice than the SE kernel for high-dimensional optimization.
>
> 4. We propose a new initialization approach that aligns conceptually with (Hvarfner et al., 2024) but differs in implementation; specifically, we select a new initialization rather than constructing a prior distribution.
>
> We will emphasize these distinctions in our paper to clarify the differences between our work and  (Hvarfner et al., 2024).
>
> >Regarding "Simpler"
>
> R2: Thank you for the detailed discussion; we agree with your points. We would like to clarify that we do **not** claim or imply that our initialization strategy is superior to constructing a prior. By "simpler," we mean that (1) it is more straightforward to implement, and (2) it avoids the need to design a distribution, which is often a nontrivial intellectual process. We are happy to remove the term "simpler" to prevent any potential confusion or debate.

---

> > ### Author Response · Authors · 2024-11-25
> >
> > Dear Reviewer,
> >
> > Thank you for your time and thoughtful feedback. As the rebuttal phase is drawing to a close, we would appreciate any further comments you may have and would like to know if we have addressed your concerns.

---

> > ### Comment · Reviewer_VLW7 · 2024-11-25
> >
> > Thank you. I have increased my score.

---

> > > ### Author Response · Authors · 2024-11-26
> > >
> > > Thank you! We appreciate your response.

---

### Meta-Review · Area_Chair_HDkR · 2024-12-19

**Metareview:**

This is another paper demonstrating that, on some problems, Bayesian optimization without e.g. local search can perform well by simply using sqrt(d) lengthscales. The overall approach does differ from Hvarfner et al., 2024 in the use of initialization versus a prior, although I'm inclined to think that the theoretical justification of vanishing GP MLL gradients is a more interesting story here. Ultimately, it's clear that the only reason *not* to have accepted this paper would have been the obvious comparison with Hvarfner et al., but all reviewers are in unanimous agreement that this is concurrent work and there are some interesting differences.

**Additional Comments On Reviewer Discussion:**

Obviously, many of the reviewers are concerned with the comparison to Hvarfner et al., 2024. I am inclined to think that pG21's point here is reasonable: the papers do appear to have been arguably concurrent work. There were a few other concerns about the broader claims in the paper and namely the required settings for these methods to work under. These seem to have been broadly addressed.

I will say that some of the points about broader claims of functions that this works for is fair -- for example, Hvarfner et al.'s sqrt(d) lengthscale scaling performs well on only 3 of the 20+ problems in Table 2 of Miguel González-Duque et al., 2024 which recently appeared at NeurIPS. This is not a deal breaker by any means -- obviously there are many problems that are much easier with simple vanilla BO that previously assumed, but not all, and so Reviewer kZCF's point about practically relevant objective functions does stand.

---

### Decision · Program_Chairs · 2025-01-22

Accept (Oral)